

# Decorrelation scales for Arctic Ocean Hydrography.
# Part I: Amerasian Basin.

Hiroshi Sumata[1], Frank Kauker[1, 2], Michael Karcher[1, 2], Benjamin Rabe[1], Mary-Louise Timmermans[3], Axel Behrendt[1], Rüdiger Gerdes[1, 4], Ursula Schauer[1], Koji Shimada[5], Kyoung-Ho Cho[6], and Takashi Kikuchi[7]

[1]Alfred-Wegener-Institut Helmholtz-Zentrum für Polar- und Meeresforschung, Bremerhaven, Germany
[2]Ocean Atmosphere Systems, Hamburg, Germany
[3]Yale University, Connecticut, U.S.A.
[4]Jacobs University, Bremen, Germany
[5]Tokyo University of Marine Science and Technology, Tokyo, Japan
[6]Korea Polar Research Institute, Incheon, Korea
[7]Japan Agency for Marine-Earth Science and Technology, Yokosuka, Japan

*Correspondence to*: Hiroshi Sumata (hiroshi.sumata@awi.de)

**Abstract.** Any use of observational data for data assimilation requires adequate information of their representativeness in space and time. This is particularly important for sparse, non-synoptic data, which comprise the bulk of oceanic in-situ observations in the Arctic. To quantify spatial and temporal scales of temperature and salinity variations, we estimate the autocorrelation function and associated decorrelation scales for the Amerasian Basin of the Arctic Ocean. For this purpose, we compile historical measurements from 1980 to 2015. Assuming spatial and temporal homogeneity of the decorrelation scale in the basin interior (abyssal plain area), we calculate autocorrelations as a function of spatial distance and temporal lag. The examination of the functional form of autocorrelation in each depth range reveals that the autocorrelation is well described by a Gaussian function in space and time. We derive decorrelation scales of 150 ~ 200 km in space and 100 ~ 300 days in time. These scales are directly applicable to quantify the representation error, which is essential for use of ocean in-situ measurements in data assimilation. We also describe how the estimated autocorrelation function and decorrelation scale should be applied for cost function calculation in a data assimilation system.



## 1 Introduction

Any use of observational data requires assumptions, or better knowledge, about the representativeness of each measurement in space and time. This holds even more for in-situ observations from data sparse regions, such as the Arctic Ocean. Interpolation guided by the statistical properties of observed quantities can provide Arctic-wide fields, while data assimilation using comprehensive dynamical models and assimilation methods can, in addition, provide fields that are consistent with the modeled physics. Also sampling strategies have to take the knowledge of the representativeness of point measurement into account. The temporal and spatial scales, for which a single measurement is representative, depend on local dynamics, external forcing and the influence of lateral water-mass influxes. Here we make an attempt to estimate those length and time scales in the Arctic Ocean based on observational data from the period 1980 – 2015. This will be achieved by estimating the autocorrelation function and decorrelation scales of temperature and salinity.

Autocorrelation functions and associated decorrelation scales are useful measures to characterize physical phenomena occurring in the ocean [*Stammer*, 1997*; Eden* 2007]. These functions describe spatial and temporal ranges over which ocean properties coherently vary, and the scales provide a measure of the spatial and temporal extent of the variations. The functional form of the autocorrelation depends on the physical properties, the considered scales (e.g., synoptic versus mesoscale) and the area. Many studies have estimated autocorrelation functions through analysis of in-situ ocean measurements [e.g., *Meyers et al.*, 1991; *Chu et al.*, 2002; *Delcroix et al.*, 2005] and satellite observations [e.g., *Kuragano and Kamachi*, 2000; *Hosoda and Kawamura*, 2004; *Tzorti et al.*, 2016]. Generally the estimated autocorrelation functions have exponential or Gaussian form [*Molinari and Festa*, 2000]. The decorrelation scales are usually given by the *e*-folding scale of the corresponding autocorrelation functions (see *McLean* [2010] for a summary of different definitions).

Estimated decorrelation scales have been applied to a variety of ocean studies. In the context of dynamical studies, the decorrelation scale is used as a measure of the scale of prevailing phenomena, and used to relate dynamical processes with the observed signals [e.g., *Stammer*, 1997; *Ito et al.*, 2004; *Kim and Kosro*, 2013]. In optimal interpolation and objective mapping, the decorrelation scale gives a measure of influential radius of a point measurement; the autocorrelation function, together with the associated decorrelation scale, provides the weight of a point measurement on mean field estimates [*Meyers et al.*, 1991; *Chu et al.*, 1997; Davis, 1998; *Wong et al.*, 2003; *Böhme and Send*, 2005]. For observation network design, decorrelation scales are one guide to estimate optimal sampling intervals in space and time [S *printall and Meyers*, 1991; *White*, 1995; *Delcroix et al.*, 2005].

One of the prevalent and growing applications of decorrelation scales is data assimilation. Data assimilation synthesizes observed data and modeled physics based on statistical theories. This is an effective approach to fill the gap between observation and modeling studies [*Wunsch*, 2006; *Blayo et al.*, 2015]. Generally data assimilation minimizes a model - data





misfit with an assessment of errors; the autocorrelation function and the decorrelation scale are necessary for these error assessments [*Carton et al.*, 2000; *Forget and Wunsch*, 2007]. They provide a direct measure of the representation error, resulting from differences of scales represented by observation and by model (see *van Leeuwen* [2015] for a summary). In ocean data assimilation, an assessment of the representation error is particularly important, since it is generally an order of magnitude larger than the measurement (instrument) error [*Ingleby and Huddleston*, 2007].

A necessity of decorrelation scale in ocean data assimilation also comes from the sparseness of ocean measurements. An autocorrelation function is necessary to constrain locations distant from a measurement. *Li et al.* [2003] pointed out that an assimilation of sparsely distributed data into an eddy permitting model, without taking its influential radius into account, causes serious problems around the locations where the data are assimilated. Artificial eddies appear around the location of the data, since the density at the data location differs from densities at their surrounding grid points in the model. They also pointed out that the assimilated information disappears on the time scale determined by the model's local advection and diffusion. Note that this situation cannot be solved by applying advanced data assimilation techniques (e.g., 4DVar, EnKF), since the artificial eddies are dynamically consistent with the modeled physics. Autocorrelation function and decorrelation scale provide necessary information to solve such problems by imposing a spatial and temporal radius of influence of each measurement [*Forget and Wunsch*, 2007; *Zuo et al.*, 2011].

Practically, autocorrelation functions are used to define an 'observation operator' in data assimilation systems. The observation operator maps modeled variables onto observational points. If the operator is properly defined, a point measurement will constrain the model, not only at the location where measurements exist, but also in areas distant from the measurement. An implementation of such an observation operator makes it possible to fully exploit the potential of sparsely distributed measurements, and can solve problems such as reported by *Li et al.* [2003]. This is of particular importance as the ocean models used for assimilation become eddy-permitting. An additional importance of autocorrelation function is to constrain the scale of temporally varying fluctuations. Unlike the static interpolation approaches, data assimilation provides a 4-dimensional analysis field. In order to appropriately assimilate observed temporal fluctuations, the temporal scale of fluctuations should be implemented in the observation operator.

In the mid-latitude and equatorial regions, there are a number of decorrelation scale estimates [e.g., *White and Meyers*, 1982; *Chu et al.*, 1997, 2002; Deser et al., 2003; *Martins et al.*, 2015], and these have been applied for a variety of studies including data assimilation (see the papers mentioned above). On the other hand, while a few studies have examined scales of temperature and salinity variability in the Arctic Ocean [e.g., *Timmermans and Winsor*, 2013; *Marcinko et al.*, 2015], there has been no assessment of basin-wide decorrelation scales of *T/S* field to date. One reason is that sea ice cover greatly inhibits sea surface observation by remote sensing. Another reason is the sparse coverage of in-situ ocean measurements due





to the inaccessibility and the absence of an Argo float network (that has provided essential data for mid-latitude and Southern Ocean studies [e.g., *McLean*, 2010; *Reeve et al.*, 2016]). In the last decade, however, the number of observational activities has been increasing significantly, with the growing concern about the sea ice retreat and its potential impact on global

climate (see e.g., *Ortiz et al.* [2011] and references therein). In addition to the increasing number of research cruises, autonomous observation platforms (e.g., Ice Tethered Profilers (ITPs) [*Krishfield et al.*, 2008a; *Toole et al.*, 2011]) now provide data throughout a full seasonal cycle in the Arctic. The data acquired from these research activities enable us for the first time to estimate basin-wide decorrelation scales for $T$ and $S$ profiles in the Arctic Ocean.


The objective for the following study is to estimate the autocorrelation functions and decorrelation scales of temperature and salinity in the Arctic Ocean at different depth. Few modelling studies have focused on applications of ocean in-situ measurements in the Arctic, due to the absence of comprehensive historical archives and representation error estimates. Only the climatology (PHC3.0: *Steele et al.* [2001]) has been widely applied for model validation [e.g., *Ilıcak et al.*, 2016]. In

recent years, however, assimilations of in-situ measurements in the Arctic Ocean have started [*Panteleev et al.*, 2004, 2007; *Nguyen et al.*, 2011; *Zuo et al.*, 2011; *Sakov et al.*, 2012]. To promote and enhance the ongoing ocean data assimilations, archiving historical measurements and estimating decorrelation scales are indispensable. To achieve the objective of the present study, 1) we compile historical observations of temperature and salinity in the Arctic Ocean, 2) construct a background mean field necessary for the decorrelation scale estimate, 3) examine the functional form of autocorrelation in

temporal- and spatial-lag space, and finally 4) provide an autocorrelation function, decorrelation scales and representation error covariance, which are directly applicable to error assessment in ocean data assimilation. Note that the estimation of the autocorrelation quantifies basin-scale variability. Smaller scale variability (e.g., mesoscale eddies on the deformation scale [*Zhao et al.*, 2014]) remains unresolved and is an intrinsic part of the autocorrelation function. The study area is the Amerasian Basin. As will be described in section 3, the second step mentioned above requires a different approach for other

regions of the Arctic Ocean. The vertical depth range of the analysis is limited to be between 0 m to 400 m depth due to data availability.

The rest of the paper is organized as follows: section 2 describes the compilation of historical data and quality control procedures applied prior to the analysis. Section 3 describes the background temperature and salinity field construction and

trend analyses. Section 4 describes examination of 2-dimensional autocorrelation functions in spatial- and temporal-lag space, and provides decorrelation scale and error covariance estimates. Section 5 gives conclusions.



## 2 Data

### 2.1 Compilation of historical data

Since there is no comprehensive in-situ ocean data archive for the Arctic, we compile historical temperature and salinity measurements with the objective not only to use the data for the present decorrelation scale estimate, but also to prepare an archive for future applications in model validation and data assimilation. Since the existing archived data from the Arctic Ocean are widely dispersed in various datasets with different formats, we compile these data into one archive with a standard format focusing on the Arctic and northern North Atlantic Ocean (Table 1). The original data (Table 1) were acquired from

various observational platforms (e.g., research vessels, moorings, ITPs and Argo floats) by conductivity temperature depth (CTD) sensors and expendable CTDs (XCTDs). The archiving effort of this study originates from the data compilation described by *Rabe et al.,* [2011, 2014] and *Somavilla et al.,* [2013], and is ongoing, thanks to support from many oceanographers. The archived data will be available on line (https://www.pangaea.de) after a profile-based thorough quality check (except those data which require additional consent from data providers).


The archived information for each measurement profile are, cruise name, station number, data type, time stamp, geographical location, bottom depth (if available), measurement depth (pressure is converted to depth by the method described by *Saunders* [1981]), temperature, salinity, data quality information provided in the original dataset (if available), and data source information. The spatial coverage of the archived data ranges from 45ºN to the pole on the Atlantic Ocean side and

from 64ºN (Bering Strait) to the pole on the Pacific Ocean side. The temporal coverage is from 1980 to 2015. Fig. 1 shows an example of the spatial distribution of the archived data (0-20m depth range, north of 64ºN) for the entire period. The archived data cover the entire Arctic and northern North Atlantic oceans, while the biggest data gaps are on the East Siberian Shelf and north of the Canadian Archipelago. A basic quality check is applied to the archived data before the duplication checks and statistical screening, described in the following subsections. The basic quality check is composed of 1)

bathymetric test using the merged IBCAO/ETOPO5 [*Jakobsson et al.*, 2012] with a tolerance of 20 m, 2) a valid range test for temperature ($-2.2 \,^\circ\mathrm{C} < T < 30.0 \,^\circ\mathrm{C}$) and salinity ($0 \, \mathrm{psu} < S < 40.0 \, \mathrm{psu}$), and 3) a vertical stability test. The bathymetric test is applied to remove data with inconsistent geographic locations (i.e., either on land, or indicating profile information at depths deeper than the sea floor at their location). This test excluded a number of erroneous profiles with position errors. If a data point violates one of the criteria, it is removed from the archive.


### 2.2 Duplication check

Since data obtained from various sources are prone to duplication issues, it is necessary to identify and remove duplicated data from the archive. A number of past studies, which compiled large oceanographic datasets, have suggested various automated procedures to deal with duplicate profiles [e.g., *Ingleby and Huddleston*, 2007; *Gronell and Wijffels*, 2008; *Good*

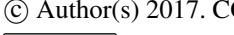



*et al.*, 2013]. In this study, we apply a simple duplication check algorithm suitable for the present application. Since we are concerned only with basin-scale variability in this analysis, we count profiles that have small spatial and temporal separations as duplicates. The threshold applied for time difference between profiles is 1 day (date coincidence) and that applied for geographical location difference is 0.05 degree in longitude and 0.01 degree in latitude, respectively; to account for the effect of convergence of meridians toward the pole, a threshold of 2 km separation is also applied. If duplication is

found (i.e., both temporal and spatial separation conditions above are met), the profiles are flagged. The profile with the highest reliability according to the data provider's own quality control is retained. For example, if we directly obtain data from PIs who have already applied their own quality control procedure, we give the data higher priority than that from other data archives (e.g., World Ocean Database 2013). The final duplication checked archive is used as input for the statistical screening described below.


### 2.3 Statistical screening

Since the archive contains a number of data that have not been quality controlled, we apply an additional quality control procedure (QC) before our analyses. The QC is composed of 2 steps: the first step is a grid-based screening; the second step is an area-based screening. Both steps are based on statistics of the data samples in discretized depth ranges. We divide the

vertical profiles of temperature ($T$) and salinity ($S$) measurements into 50 depth bins (from 20 m interval near the sea surface to 200 m interval in the deep ocean, Fig. 2a). If there are more than two measurements for a certain depth range from one profile, the measurement values ($T$ and $S$) are averaged. The statistics are calculated and applied in each depth range separately.

First we apply a grid-based screening. The grid-based screening takes the difference in statistics (mean and standard deviation) in different locations into account. We define 111 km × 111 km (corresponding to 1° × 1° at the Equator) grid cells over the entire archive domain. The mean ($\mu$) and standard deviation ($\sigma$) of $T$ and $S$ on each grid cell and in each depth range are calculated from the data within the surrounding 555 km × 555 km (5° × 5°) area. $T$ and S values outside the 5-times standard deviation ($\mu \pm 5\sigma$) on each grid cell are removed from the archive (the procedure is repeated twice).


Second we apply an area-based screening for the data deeper than 750 m depth. In this step we apply more rigorous statistics calculated from the entire basin and shelf area. This step is necessary to remove problematic data in data sparse areas and in data sparse depth ranges, since the grid-based screening cannot provide good statistics in these areas due to the small sample size (no ITP data below 750 m). We classify the archived data into 6 sub-domains based on the characteristics of dynamical

regimes [*Nurser and Bacon*, 2014]: 1) Amerasian Basin, 2) Amerasian shelf and shelf slope, 3) Siberian shelf and shelf slope, 4) Eurasian Basin, 5) Barents and Kara Sea including their shelf slopes and 6) Nordic Seas (Fig. 2b). Mean and



standard deviation are calculated in individual sub-domains. Then data outside the 5 times standard deviation (μ±5σ) are removed (repeated twice). In this paper, we focus only on the results for the Amerasian Basin; regions 2 – 5 are considered in a separate analysis.


The result of the statistical screening in the Amerasian Basin is shown in Fig. 3. The combined statistical screening successfully removes spurious data in deep depth ranges, while retaining the relatively larger variability in shallow depth ranges. After the combined statistical screening, the vertically discretized data are used for the analyses in the following section.


**3 Construction of the background mean field**

In this section we describe the construction of a background mean field of $T$ and $S$, which represents the basin-wide climatology in the Amerasian Basin. The background mean fields will be used to calculate anomaly fields necessary for the

decorrelation scale estimates. For the construction of the background mean field, we first examine the functional form and spatial scale of the mean field variation (sec. 3.1). Second we apply the derived functional form and scale for the background mean field construction (sec. 3.2). The temporal linear trends of $T$ and $S$ are also examined to account for the effect of a long-term temporal change of the mean field (sec. 3.3).

**3.1 Spatial scale of variation**

To derive the scale for the background field construction, we examine the spatial scale of variation in each depth range (The vertical layers defined in Fig. 2b are used throughout this study, to provide decorrelation scales directly applicable to data assimilation systems using z-coordinate systems). In this estimation we assume isotropy and homogeneity of the spatial scale of variation in a basin. These assumptions are valid if 1) planetary- and 2) topographic-$\beta$ effects do not dominate in a basin,

and 3) no dominant oceanic structure extends toward one specific direction. The first and second conditions are satisfied in the high-latitude Amerasian Basin (small planetary-$\beta$ effect) away from marginal shelf slopes, where a large topographic-$\beta$ effect is expected. The third condition is also satisfied in the deep Amerasian Basin, although not necessarily in other sectors of the Arctic Ocean and Nordic Seas. For example, in the Eurasian Basin there is a prominent extension of the frontal structure along the shelf slope associated with the warm Atlantic water inflow [*Anderson et al.*, 1994; *Rudels et al.*, 2013].

The location of the front is not necessarily trapped over the shelf slope, but can be detached from the slope [*Jones*, 2001]. Further, in the Nordic Seas there are meridionally-extending dominant current systems, i.e., the East Greenland current, Norwegian Atlantic Current and West Spitsbergen Current [*Hopkins*, 1991]. These features require a scale examination that



takes a spatial anisotropy into account; a different approach for scale estimation will be applied to the Eurasian Basin in a forthcoming paper. For our purposes here, the Amerasian Basin is defined by the area where total water depth is deeper than

1000 m. This definition excludes the area affected by coastal currents and topographically-trapped flows (associated with the submarine Northwind Ridge, for example).

To estimate the spatial scale of variation we introduce a structure function [*Davis et al.*, 2008; *Todd et al.*, 2013] with the assumption of spatial and temporal isotropy of variation,


$$S_{x,t} = \left\langle \left[ T(x_0 + x, t_0 + t) - T(x_0, t_0) \right]^2 \right\rangle ,$$

(1)

where $x$ and $t$ are the spatial and temporal separation from location $x_0$ and time $t_0$, $T$ is the observed property (in this case,

either $T$ or $S$), and $\langle \cdot \rangle$ is the averaging operator over space and time. The structure function, $S_{x,t}$, gives the mean square

difference between two measurements as a function of spatial and temporal separations. It was initially introduced by *Kolmogorov* [1941] to provide a statistical description of a field without specifying the mean and variance of the field. This is an appropriate approach for the present purpose, since we do not have a priori information regarding the statistics of the background field. We calculate the structure function from all available data in the Amerasian Basin (all depth bins shallower than 400 m),


$$S_{x,t} = N^{-1} \sum_{i=1}^{N} \Delta T_i(x,t)^2 ,$$

(2)

where $N$ is the number of available data pairs, the spatial and temporal separation of which are $x$ and $t$, and $\Delta T_i(x, t)$ is the difference of observed values of the $i$-th pair. We introduce a function $f$, which measures the normalized root-mean-square

difference (RMSD) of any two measurements,

$$f(x,t) = 1 - \left( \frac{S_{x,t}}{S_{bg}} \right)^{\frac{1}{2}} ,$$

(3)

where $S_{bg}$ is defined by all the possible combinations of available data in the basin in a certain depth range,




$$S_{bg} = \frac{2}{M(M-1)} \sum_{i=1}^{M-1} \sum_{j=i+1}^{M} (T_i - T_j)^2 \quad , \tag{4}$$

and $M$ is the number of all available data. $S_{bg}$ is a measure of the size of basin-wide and long-term variations, i.e., we introduce it as the 'background' mean squared difference used to normalize $S_{x,t}$.


The function $f$ in Eq. (3) is a unit-less measure of RMSD between two measurements as a function of spatial and temporal separation. If $S_{x,t} \sim S_{bg}$, i.e., the mean difference between two measurements with $(x, t)$ separation is comparable to those of 'large' distance measurement pairs, then $f \sim 0$. This indicates that no coherent structure exists between data with $(x, t)$ separation. If $S_{x,t} \ll S_{bg}$, i.e., the mean difference between measurements with $(x, t)$ separation is sufficiently small

compared to that between sufficiently distant data pairs, then $f \rightarrow 1$. This indicates a strong coherence exists between the data with $(x, t)$ separation (ultimately $f = 1$, if the spatial and temporal separation are exactly zero). Note that the function $f$ is not an autocorrelation function, although it has similar properties (e.g., decays from 1 to 0 for spatial and temporal separation from zero to infinity). The function $f$ measures the scale of the coherent structure of the mean field, whereas an autocorrelation function measures the scale of coherent variation of anomalies. A structure function $S$ can be directly

related to an autocorrelation function, if we can define $S$ by the anomaly from the mean field [e.g., *Gandin* 1965; *Molinari and Festa*, 2000]. Since we have no a priori statistical information regarding the mean field, we cannot relate the structure function $S$ with the autocorrelation in our case. The correspondence to the geostatistical approach is given in Appendix A.

In order to examine the functional form of $f$, we construct data pairs from all possible combinations of data in each depth range, classify the pairs into 50 × 36 bins (50 bins for spatial separation with 10 km interval and 36 bins for temporal

separation with 10-day interval), and calculate $f$ in respective bins. Examples of the functional form of $f$ for $T$ and $S$ in spatial- and temporal-separation space are shown in Fig. 4. Small separation gives large $f$, while $f \sim 0$ when the separation is sufficiently large. Note that $f$ decays with an increase in temporal separation in shallow depth ranges with a time scale of approximately 90 - 120 days (Fig. 4a, b), while $f$ is relatively insensitive to temporal separations at depths deeper than 80 m (Fig. 4c, d), which is a manifestation of the seasonality. This seasonality is taken into account to estimate the background

mean field in sec. 3.2. Note that we limit our analysis here to consider only the upper water column, from 0 m to 400 m depth as uncertainties in the uncalibrated ("level-2") ITP salinity data are comparable to the temporal and spatial variability of salinity in the Amerasian Basin below 500 m (see Appendix B).



To closely examine the functional form of *f*, we calculate the temporal (0-90 days) average of *f* in respective depth ranges. A
survey of the 2-dimensional functional form over all depth ranges (shallower than 400 m) revealed that 90-days is a
reasonable choice to account for seasonal variation (not shown). Fig. 5 shows the 90-day averaged functional form of *f* in
different depth ranges (thin-dotted lines) and the average for all depth ranges (0-400 m; thick-dotted black line). Although
the scale of variation varies with depth, the functional form of which can be reasonably approximated by a Gaussian function
(thick-solid blue line). Note that *f* does not come close to 1, even if the spatial separation nears 0 km, because the present
examination excludes self combination of data (i.e., $\Delta T(0, 0) = 0$), deals with 0 - 90 day average, and does not resolve
mesoscale fluctuations smaller than 10 km scale (the spatial separation of the bin).

The e-folding scales of the fitted Gaussian function for *T* and *S* are summarized in Fig. 6. The *T* profile (dashed black line)
exhibits a large spatial scale of variation (~ 200 km) near the sea surface, indicating the effect of the large-scale thermal
forcing at the sea surface. The *T* profile deeper than 100 m depth is nearly constant (120 ~ 150 km). The salinity profile
(solid blue line), on the other hand, exhibits nearly constant scale (130 ~ 150 km) from the sea surface to 400 m depth,
indicating small contributions from large-scale surface salinity fluxes at the sea surface. We apply the *e*-folding scale of each
depth level and the Gaussian function to estimate the background mean field.

## 3.2 Background mean field

To take the seasonal variation into account, we divide the observed data into 4 seasons (January - March, April - Jun, July -
September, and October - December), and construct the background mean *T* and *S* fields in each season. This is supported by
the fact that the temporal *e*-folding scale is approximately 90 days in shallow layers (Fig. 4a, b) and even longer in the
deeper layers. The background field is derived by applying a spatial Gaussian filter with an *e*-folding scale given by the
spatial scale of variation in each depth range (Fig. 6). The background field for $T_i$ is given by

$$\bar{T}_i = \sum_{n=1}^{N} W'_n T_n ,$$

(5)

where *N* is the number of measurements, whose distance from the *i*-th measurement ($T_i$) is less than 3 times the *e*-folding
scale (i.e., $\| x_i - x_n \| < 3L$, see below), $W'_n$ is the normalized weighting function for the *n*-th data point, and $T_n$ is the *n*-th
measurement surrounding the *i*-th measurement. The normalized weighting function $W'_n$ is given by

$$W'_n = \left( \sum_{n=1}^{N} W_n \right)^{-1} W_n ,$$

(6)





where $W_n$ is the Gaussian weighting function:

$$W_n = \exp\left[-\left(\frac{\left\|\boldsymbol{x}_i - \boldsymbol{x}_n\right\|}{L(z)}\right)^2\right],$$   (7)

where $\boldsymbol{x}_i$ and $\boldsymbol{x}_n$ are the geographical location of $T_i$ and $T_n$, respectively, and $L(z)$ is the $e$-folding scale of the Gaussian filter as

a function of depth (Fig. 6). An example of the derived background field for $T$ and $S$ in summer is shown in Fig. 7. The field captures a warm and fresh water mass distribution in the Canada Basin and its smooth transition toward cold and saline water in the northeastern Amerasian Basin. For the anomaly field calculation, we require the background field at the locations where observational data exist. Therefore we do not apply any spatial and/or temporal interpolations even in data-sparse seasons (winter and spring).


### 3.3 Temporal trend

For the present anomaly derivation, we also take the temporal trend from 1980 to 2015 into account. The trend is estimated in each 111 km × 111 km grid cell, in each depth range, and in each season (Mann-Kendall rank statistics [*Kendall*, 1938] with a significance level of 5%). Fig. 8 shows representative $T$ and $S$ trends in the 60 - 80 m depth range in summer and the

corresponding average time series for those grid cells for which the trend is statistically significant. A warming (~0.5 deg/decade) and freshening (~0.5 psu/decade) trend in the Canada Basin is evident in this depth range. The freshening trend extends from the sea surface to 400 m depth without significant change in spatial pattern, whereas the $T$ trend changes sign and spatial pattern with depth. A positive $T$ trend is observed in the depth range from 0 m to 160 m depth (i.e., through the Pacific-water/upper halocline layers), whereas a decreasing $T$ trend is observed in the 200 - 400 m depth (lower

halocline/Atlantic-water layer) in the central Canada Basin (Fig. 8c, blue line) after 2002. A positive trend is observed in 250 - 400 m depth (Atlantic-water layer) along the southern perimeter of the Canada Basin (Fig. 8c, black line).

The warming and freshening trend in the Pacific-water layer has already been reported by many studies [e.g., *Proshutinsky et al.*, 2009; *Jackson et al.*, 2010; *Giles et al.*, 2012; *Timmermans et al.*, 2014]. The cooling trend in the central Canada basin

and the warming trend along its southern perimeter is a consequence of deepening of the warm Atlantic Water in the central basin and concurrent upwelling of warm Atlantic Water at the boundaries, a manifestation of an intensification of the anticyclonic Beaufort Gyre in recent years [e.g., *McLaughlin et al.*, 2009; *Karcher et al.*, 2012; *Zhong and Zhao*, 2014]. Note that this trend estimate does not necessarily represent the long-term trend for 1980 - 2015. The estimated trends in each 111





km × 111 km area represent only the data-covered period. Although similar trends can be found in other seasons (from

winter to spring), they are not statistically significant.

The temporal trend in each location is used to define a time-varying background field. Since the temporal distribution of the

archived data is not spatially uniform, the representative time (i.e., the time that the temporal mean value represents) of the

background field $\bar{T}_i$ varies with space. The representative time is used as a tie point (offset) to connect the mean and trend.

Taking the effect of the representative time into account, the time-varying background field for $T_i$ is defined by

$$\widetilde{T}_i = a(\mathbf{x})[t - t_{rep}(\mathbf{x})] + \bar{T}_i \,, \tag{8}$$

where $a(\mathbf{x})$ is the temporal trend at location $\mathbf{x}$, $t$ is the time, $t_{rep}(\mathbf{x})$ is the representative time of the background mean field $\bar{T}_i$

at location $\mathbf{x}$. We calculate the representative time in each 111 km × 111 km area by the average of measurement times of all

the data contained in the corresponding area, and apply it to define the time-varying background field. For the area where no

trend can be deduced, we apply a constant background field, $\widetilde{T}_i = \bar{T}_i$ .

## 4 Decorrelation scale

### 4.1 Autocorrelation function

Decorrelation scales used in oceanographic studies are generally defined by an *e*-folding scale of an autocorrelation function,

which has a Gaussian or exponential functional form [*Molinari and Festa*, 2000]. Practically, the autocorrelation functions

are obtained from a series of autocorrelations estimated by differently-lagged points [e.g., *White and Meyers*, 1982; *Meyers*

*et al.*, 1991]. An autocorrelation for $\Delta l$ lag is given by


$$\rho_{l,l+\Delta l} = \frac{cov(\mathbf{T}_l , \mathbf{T}_{l+\Delta l})}{\sqrt{var(\mathbf{T}_l) \cdot var(\mathbf{T}_{l+\Delta l})}} \,, \tag{9}$$

where $cov(\mathbf{T}_l, \mathbf{T}_{l+\Delta l})$ is an autocovariance between two data series $\mathbf{T}_l$ and $\mathbf{T}_{l+\Delta l}$, the temporal and/or spatial lag between which

are $\Delta l$, and $var(\mathbf{T}_l)$ and $var(\mathbf{T}_{l+\Delta l})$ are the variances of $\mathbf{T}_l$ and $\mathbf{T}_{l+\Delta l}$, respectively. We assume isotropy and homogeneity of the

autocorrelation in the Amerasian Basin, supported by the fact of weak planetary-$\beta$ effect in polar regions and the

homogeneity of the Rossby radius in the Amerasian Basin [*Nurser and Bacon*, 2014; *Zao et al.*, 2014]. These assumptions



enable us to calculate the autocorrelation from data series, which are composed of data pairs having the same temporal and spatial lag $\Delta l$, but coming from different locations in the basin and from different times [e.g., *Sprintall and Meyers*, 1991; *Chu et al.*, 1997, 2002], i.e.,


$$\rho_{\Delta l} = \frac{\sum_{n=1}^{N} T'_n \hat{T}'_n}{\sqrt{\sum_{n=1}^{N} \left(T'_n\right)^2 \cdot \sum_{n=1}^{N} \left(\hat{T}'_n\right)^2}}, \qquad (10)$$

where $N$ is the number of data pairs, the spatial and temporal lag between which are $\Delta l$, $T'_n$ is the anomaly value of the $n$-th data, $\hat{T}'_n$ is the anomaly value of the paired data which locates $\Delta l$-lagged point from $T'_n$.


The anomaly data set $T'$ is defined by subtracting the time-varying background field $\widetilde{T}$ from the observed data $T$. Each anomaly data of the set is paired with the other anomalies to construct a set of anomaly data pairs, which consists of all possible combinations of two anomaly data. The data pairs are classified into discretized bins, according to the spatial and temporal lag of the paired data (50 spatial bins with a 10-km interval and 73 temporal bins with a 5-day interval, i.e., the examination window is 500 km-lag × 365 days-lag). Each bin has a sufficient number of data pairs to calculate an autocorrelation ($N > O(10^3)$, see Fig. 9a). Fig. 9, panels b and c, show examples of the autocorrelation functions for $T$ and $S$ in the 40 - 60 m depth range. There is a clear decrease of autocorrelation with increasing spatial and temporal lag, although with some variability about this relationship.

Temporal and spatial averages of the autocorrelation are calculated to identify its functional form by fitting a suitable empirical function. Fig. 10a and b show the temporal average of the spatial autocorrelation functions of $T$ and $S$ for different depth ranges. To account for the effect of differences of temporal autocorrelation scales in different depth ranges, we define the temporal average by 0 - 30 days lag in shallow levels (0 - 140 m depth range), and by 0 - 60 days lag in deeper levels (below 140 m). The functions generally show their highest values at zero spatial lag, with decreasing values as the spatial lag increases. Some functions exhibit a second peak around a spatial lag of 200 - 300 km. We examine the relation between the second peaks and associated background mean field of $T$ and $S$ in different depth ranges, and find that the peaks derive from the circular $T$ and/or $S$ structure of the Beaufort Gyre (see Appendix C). Since the Beaufort Gyre is characterized by bowl-shaped isosurfaces of $T$ and $S$ associated with surface downward Ekman pumping, coherent variation of the isosurfaces gives rise to the second peak. To eliminate the effect of the second peak for our scale estimate, we use the autocorrelation functions just for a spatial lag of 0 - 150 km to compute a fitting function. We tested exponential and Gaussian functions for the fitting,



and found that the Gaussian function is generally suitable to represent the observationally-derived spatial autocorrelations (Fig. 10c, d).

The temporal autocorrelation is also examined by taking spatial-lag averages (0 - 20 km) of the 2-dimensional
autocorrelations of $T$ and $S$. Fig. 11a, b show the averaged temporal autocorrelation functions in various depth ranges. The functions show their highest values at zero temporal lag and a reduction towards large temporal lags, whereas the functions from many depth ranges clearly exhibit an annual cycle. Since the seasonal variability of the background field is already taken into account (sec. 3.2), the annual cycle found in the temporal autocorrelations indicates the effect of persistent atmospheric forcing, the time scale of which is longer than 1 year (e.g., Arctic Oscillation [*Thompson and Wallace*,1998],
North Atlantic Oscillation [*Hurrell*, 1995; *Wallace*, 2000]), and/or spin-up/down process of gyre-scale circulation, the time scale of which is estimated as 3 - 4 years [*Yoshizawa* et al., 2015]. To remove the effect of the annual cycle found in Fig 11a, b, we use the autocorrelation functions from 0 - 200 days temporal lag to find a fitting function for the temporal autocorrelation. We again tested exponential and Gaussian forms for the fitting, and found that the Gaussian functions are suitable to represent the form of the temporal autocorrelation functions (Fig. 11c, d).

### 4.2 Decorrelation scale

The spatial and temporal decorrelation scales of $T$ and $S$ are derived from the *e*-folding scales of the fitted spatial- and temporal-autocorrelation functions in the respective depth ranges. The spatial autocorrelation function is represented by the Gaussian form,

$$\rho_s = A_s \cdot \exp\left[-\left(\frac{x}{d_s}\right)^2\right],$$

(11)

where $A_s$ is the autocorrelation at zero-spatial lag, $x$ is a spatial lag, and $d_s$ is the spatial decorrelation scale. The temporal autocorrelation function has the same formula, but exchanging $A_s$ for $A_t$, $x$ for $t$, and $d_s$ for $d_t$, where $A_t$, $t$, and $d_t$ are the
autocorrelation at zero-temporal lag, temporal lag, and temporal decorrelation scale, respectively. The autocorrelation at zero temporal and spatial lag ($A_s$ and $A_t$) represents the effect of unresolved fluctuations, which have a scale smaller than the resolution of the present analysis at 10 km resolution in space and 5 days resolution in time ($1 - A_s$ represents the magnitude of unresolved fluctuations relative to the basin-scale fluctuations). The effect of mesoscale eddies with the scale of the deformation radius (order 10 km horizontally) are described by this parameter.





Fig. 12 summarizes the vertical profiles of the spatial and temporal decorrelation scales ($d_s$ and $d_t$) of $T$ and $S$ with the associated parameters for zero-lag autocorrelations ($A_s$ and $A_t$). The zero-lag autocorrelations (Fig. 12a, c) show smaller values (0.6 - 0.7) in the upper 100 m depth range, indicating active mesoscale processes (e.g., eddy activity observed in the Pacific-water layer [*e.g., Zhao et al.*, 2014]). The zero-lag autocorrelations for spatial (Fig. 12a) and temporal lags (Fig. 12 c)
exhibit similar profiles, confirming the appropriateness of the spatial and temporal averages used for the functional form examinations. The vertical profiles of the decorrelation scale (Fig. 12 b, d) indicate an influence of the sea surface boundary condition at shallow levels. The spatial decorrelation scale near the sea surface (~ 200 km) is larger than it is in deeper layers (~ 150 km), as a consequence of the direct influence of the atmosphere and sea ice, the spatial scale of which is larger than the scale of intrinsic ocean processes. The temporal decorrelation scale near the surface (100 ~ 150 days), on the other hand,
is shorter than that of the deeper layers (200 ~ 300 days), due to the effect of short-timescale variation of the atmospheric field and associated sea ice motion.

Note that the $T$ and $S$ profiles exhibit similar vertical profiles in the depth range shallower than 250 m, while discrepancies stand out in levels deeper than 250 m (Fig. 12b, d). This may be due to small calibration errors associated with our use of
ITP level-2 (i.e., not the fully calibrated level-3) data [see Krishfield et al., 2008b, Johnson et al., 2007]. In order to incorporate as many data as possible, we have included all available ITP level-2 data, where level-3 data are not yet available. This strategy is beneficial for scale estimation of temperature (ITP level-2 temperature data have the same accuracy as level-3 data, within ±0.001 °C) in the entire depth range and salinity shallower than 250 m depth. On the other hand, since salinity variability decreases with depth (Fig. 3b), the uncalibrated ITP level-2 salinity data may yield non-
negligible spurious variation at levels deeper than 250 m, which may deteriorate the accuracy of the scale estimates for salinity in this depth range.

### 4.3 Error covariance

The autocorrelation function derived in sec. 4.1 can be related to an error covariance by Eq. (9). Since the variance in Eq. (9)
used to normalize the covariance does not depend on spatial and/or temporal separation in principle (see the assumption in sec. 4.1), it can be represented by a variance calculated from all the data in the Amerasian Basin. Therefore, the error covariance associated with the representation error is given by a function of spatial and temporal separation, $x$ and $t$,

$$cov(x,t) = \rho(x,t) \cdot var_{bg} \quad , \tag{12}$$


where $\rho(x, t)$ is the autocorrelation function, and $var_{bg}$ is the background mean variance defined by



$$var_{bg} = \frac{1}{M} \sum_{i=1}^{M} \left( \widetilde{T}_i - T_i \right)^2 \quad . \tag{13}$$

460 The vertical profiles of $var_{bg}$ for $T$ and $S$ are shown in Fig. 13.

**5 Conclusion**

We examined spatial and temporal scales of $T$ and $S$ anomalies from the mean fields in the Amerasian Basin. To provide scales describing the anomalies, we examined the autocorrelation of $T$ and $S$ measurements and calculated spatial and

465 temporal decorrelation scales. Historical $T$ and $S$ measurements in the Arctic and northern North Atlantic oceans were compiled for this study and for future applications to Arctic ocean data assimilations. The resulting quality controlled archive was used to construct a background mean field, from which anomaly fields were derived. By assuming spatial and temporal homogeneity of the autocorrelation function in the basin interior, we calculated autocorrelations as a function of spatial and temporal lags. The examination revealed that the autocorrelation function can be well described by a Gaussian function in

470 space and time. The spatial and temporal decorrelation scales were estimated to be 150 ~ 200 km in space and 100 ~ 300 days in time (*e*-folding scales of the autocorrelation function). The spatial decorrelation scale is relatively large near the sea surface, while the temporal scale is relatively small near the surface. Mesoscale fluctuations, with scales smaller than 10 km and shorter than 5 days, are represented by the zero-lag autocorrelation. The zero-lag autocorrelation should be re-examined in future work to describe the autocorrelation smaller than the Rossby radius by fully exploiting ITP data.


The estimated function and the scales, together with the associated error covariance, are directly applicable to model - observation misfit calculation in data assimilation systems, which intend to assimilate a spatially and temporally varying field. A cost function measuring the model - observation misfit is given by

$$J = \frac{1}{2} \left[ \boldsymbol{d} - \boldsymbol{H}(\boldsymbol{m}) \right]^T \mathbf{R}^{-1} \left[ \boldsymbol{d} - \boldsymbol{H}(\boldsymbol{m}) \right] \quad , \tag{14}$$

where $\boldsymbol{d}$ is the data vector, $\boldsymbol{m}$ is the model vector, $\boldsymbol{H}$ is the observation operator, $\mathbf{R}$ is the observation error covariance matrix. The current study gives the descriptive form of $\boldsymbol{H}$ and $\mathbf{R}$. An observation operator, $\boldsymbol{H}$, which takes spatial and temporal representativeness of each measurement into account, is given as follows;




$$H_i(\boldsymbol{m}) = \frac{\sum_{j=1}^{M} m_j \rho(x_{ij}, t_{ij})}{\sum_{j=1}^{M} \rho(x_{ij}, t_{ij})} \quad , \tag{15}$$

where $i$ refers to the $i$-th in-situ measurement, $j$ refers to the modeled variable at the $j$-th model grid point, $\rho$ is the autocorrelation between $(x, t)$-distant locations, $x_{ij}$ and $t_{ij}$ are the spatial and temporal separation between the $i$-th

measurement and the $j$-th model grid point. The operator $H_i(\boldsymbol{m})$ maps the model field $\boldsymbol{m}$ to the $i$-th measurement location (in space and time), in accordance with the influence of the measurement. We can describe the autocorrelation function $\rho$ by the results shown in sec. 4.1 and 4.2 in the following formula,

$$\rho(x, t) = A \cdot \exp\left[ -\left(\frac{x}{d_s}\right)^2 - \left(\frac{t}{d_t}\right)^2 \right] \quad , \tag{16}$$


where $A$ is the autocorrelation between zero-lag locations ($x < 10$ km and $t < 5$ days) representing the contributions from unresolved-scale fluctuations (Fig. 12a), $d_s$ and $d_t$ are the spatial and temporal decorrelation scales (Fig. 12b, d), respectively. This formula provides the representation error of a point measurement at $(x, t)$-distant locations. Note that the current

formula enables us to quantify errors of modeled $T$ and $S$ not only at the location where the measurements exist, but also at the locations distant from the measurements. The present study also provides error covariance matrix $\mathbf{R}$ associated with the representation error. The representation error covariance between the $i$-th and the $i'$-th measurement is

$$cov(i, i') = \rho(x_{ii'}, t_{ii'}) \cdot var_{bg} \quad , \tag{17}$$


where $\rho(x_{ii'}, t_{ii'})$ is the autocorrelation between $i$-th and $i'$-th measurement, the spatial and temporal separation between which are given by $x_{ii'}$ and $t_{ii'}$, and $var_{bg}$ is the background error variance given as a function of depth (Fig. 13). As summarized here, the current study provides a full descriptive formula to exploit ocean in-situ measurements in the Amerasian Basin for a model - observation misfit calculation.


The present scale estimates pose a requirement from a basin-scale data assimilation on a sampling strategy. Static interpolation approaches (e.g., optimal interpolation [Gandin, 1965; Reynolds and Smith, 1994], objective mapping [*Wong et al.*, 2003; *Böhme and Send*, 2005; *Böhme et al.*, 2008], and Data-Interpolating Variational Analysis [*Troupin et al., 2010, 2012; Korablev, 2014*] ) exploit statistical information of data to derive a mean analysis field. Data assimilation approaches,



in addition, exploit modeled physics and provide temporally and spatially varying 4-dimensional analysis fields. The former approaches need a scale representing the mean field, while the latter, in addition, needs spatial and temporal scales representing the anomaly field to fully exploit the information embedded in in-situ data. For Arctic Ocean studies, statistical interpolation has been using decorrelation scales of 300 ~ 500 km (*Steele et al.*, [2001], *Proshutinsky et al.*, [2009], *Rabe et al.* [2011, 2014]), while the present study suggests the necessity of a smaller measurement interval (150 ~ 200 km in space

and 100 ~ 300 days in time) to describe the anomaly field by a basin-scale data assimilation.

Further studies are necessary to interpret the decorrelation scale of $S$ and $T$ in the context of ocean dynamics and relate it to the hydrographic features in the Amerasian Basin. The scale of ocean variability is governed by external forcings and by various physical processes in the ocean. The local dynamic response to local external forcing (i.e., vertical normal mode in

response to basin-scale wind stress curl [*Pedlosky*, 1987; *Olbers et al.*, 2012]) is one very likely mechanism to explain the shape of the vertical profile of the scale. Near the sea surface the decorrelation scales should be examined in relation to the scale of atmosphere and sea ice variability [*Walsh*, 1978; *Walsh and Chapman*, 1990], and the dynamical processes governing the mixed layer [*Peralta-Ferriz and Woodgate*, 2015]. The effect of remote forcing is another important issue to be examined. Advection of anomalous water masses introduce scales governed by mechanisms outside of the basin and/or

shelf-basin interaction, such as the inflow of anomalous Pacific Water into the deep basin [*Steele et al.*, 2004; *Itoh et al.*, 2012], its modification processes on the shelf [*Pickart et al.*, 2005, *Woodgate et al.*, 2005], the advection of anomalous Atlantic Water [*McLaughlin et al.*, 2009; *Karcher et al.*, 2012], or variations of freshwater supply due to river runoff [*Lammers et al.*, 2001]. In this study we employed level surfaces, as we focus on the applicability of the decorrelation scales for model validation and data assimilation (many models use the so called z-coordinate system). For future studies which

aim at a dynamical interpretation of the decorrelation scales, an analysis in isopycnal coordinates would be a logical next step. Autocorrelation and decorrelation scale estimates for other parts of the Arctic Ocean (i.e., the Eurasian Basin, and over the shelf slopes) will be presented in forthcoming papers.

**Acknowledgement**

Funding by the Helmholtz Climate Initiative REKLIM (Regional Climate Change), a joint research project of the Helmholtz Association of German research centers (HGF) is gratefully acknowledged. This work has partly been supported by European Commission as part of FP7 project Ice, Climate, and Economics - Arctic Research on Change (ICE-ARC, Proj. Nr., 603887). We also would like to express our gratitude towards the German Federal Ministry of Education and Research (BMBF) for the support of the project "RACE II - Regional Atlantic Circulation and Global Change" (03F0729E) and

various observational efforts listed in Table 1. The data in the Amerasian Basin were collected and made available by the following research programs: Arctic Switchyard project (http://www.ldeo.columbia.edu/Switchyard), Baufort Gyre





Exploration Program based at the Woods Hole Oceanographic Institution (http://www.whoi.edu/beaufortgyre) in collaboration with researchers from Fisheries and Oceans Canada at the Institute of Ocean Sciences, the second and third Chinese National Arctic Research Expeditions [*Shi*, 2009a, 2009b], Ice-Tethered Profiler Program [*Toole at al.*, 2011;
*Krishfield et al.*, 2008] based at the Woods Hole Oceanographic Institution (http://www.whoi/edu/itp), JAMSTEC Compact Arctic Drifter (J-CAD) measurements by the North Pole Environmental Observatory Project led by University of Washington in collaboration with researchers from Japan Agency for Marine-Earth Science and Technology (JAMSTEC) [*Kikuchi et al.*, 2004], the KPDC (http://kpdc.kopri.re.kr) data archived from the project titled 'K-AOOS' (Korea Polar Research Institute, PM16040)' funded by the Ministry of Oceans and Fisheries, Korea, LOMROG 2007 Oden cruise [*Bjork,*
*G. Dr. and Gothenburg University*, 2012], Nansen and Amundsen Basins Observational System (NABOS/CAOBS) based at University of Alaska Fairbanks (http;//nabos.iarc.uaf.edu/index.php), North Pole Environmental Observatory (NPEO) [*Morison et al.*, 2002], RV Mirai cruises operated by JAMSTEC (http://www.godac.jamstec.go.jp/darwin/), Submarine Arctic Science Program (SCICEX) [*SCICEX Science Advisory Committee*, 2009], the UNCLOS 2011 program by Fisheries and Oceans Canada at the Institute of Ocean Science in collaboration with JAMSTEC [*Guéguen et al.*, 2015], and World
Ocean Database 2013 [*Boyer et. al.*, 2013]. The GFD-DENNOU library (http://dennou.gaia.h.kyoto-u.ac.jp/arch/dcl/) and Ocean Data View [*Schlitzer*, 2015] were used to draw the figures.

**Appendix A: Correspondence to the geostatistical approach**

Since data analysis softwares based on geostatistical approaches (e.g., iSATiS, SURFER) are used in oceanographic studies
in recent years, it is useful to providing a summary of the relation between the current approach and geostatistical approaches. The spatial scale of variation estimated in section 3.1 is a different notation of the variogram concept used in geostatistics. In the present formula, we normalize the variance by the sill of the variogram, and a root-squared value is considered. This is because a variogram deals with a variance (i.e., spatial scale of the *squared* difference between two measurements), while we intend to quantify the spatial scale of difference between two measurements. We also defined the
function by the value subtracted from 1, in order to obtain a function decaying to zero at infinity. This is done for mathematical convenience in order to obtain a Gaussian-like function. This is preferable for the framework of the best linear unbiased estimator (BLUE), which is constituting the basis of data assimilation theories. Since the spatial scale of variation originates from the same concept as variograms, it can be related to the terminology used in geostatistical approaches. The function $f$ (i.e., normalized root-mean-square-difference) at zero separation (Fig. 5) is

$$f\big|_{x=0} = 1 - \sqrt{\frac{2N_g}{S_{bg}}} \quad , \tag{A1}$$





where $N_g$ is a nugget of semi-variogram plot. The estimated scale (the spatial scale of variation) describes the square-root of the scale described by a variogram, although it is not easy to find an exact correspondence, since empirical functions describing the two functions may differ. If we directly translate the function $f$ into a semi-variance used to plot a semi-variogram, our formulation corresponds to an empirical semi-variance with the following form,

$$\hat{\gamma}(x) = \frac{S_{bg}}{2}\left[ Ae^{-(x/L(z))^2} - 1 \right]^2 \quad , \tag{A2}$$

where $A$ is the function $f$ value at zero separation, which is related to the nugget in Eq. (A1). Since we modeled the function $f$ by a Gaussian formula, we cannot define the '*range*' in the corresponding semi-variogram (the range goes to infinity in a Gaussian formula). After obtaining a background mean field by using the spatial scale of variation, we do not have to rely on geostatistical approaches any longer, since we can directly calculate the autocorrelation by variance and auto-covariance (Eq. 9).

**Appendix B: Error estimates of ITP level-2 data**

Woods Hole Oceanographic Institution provides ITP temperature and salinity data at different levels of processing; here we use both level-3 (final processed data) and uncalibrated level-2 data when level-3 data are not available [see *Krishfield et al.,* 2008b]. Profile-by-profile conductivity calibration (not applied to the level-2 data) accounts for conductivity sensor drift. The calibration method applied to level-3 data is to adjust the potential conductivity of each profile to the value derived from bottle-calibrated CTD stations on the deep 0.4 °C potential temperature surface [Krishfield et al., 2008b].

As a measure of the uncertainty of the uncalibrated ITP level-2 data, we estimate deviations of ITP level-2 data from the background mean field (sec 3.2). Vertical profiles of the standard deviations of $T$ and $S$ calculated from all data, from ITP level-2 data, and from all data except ITP level-2 are shown in Fig. B1. Below about 250 m depth, there is significant variance in ITP level-2 data, and this becomes even larger below 500 m depth. To avoid uncertainty associated with uncalibrated ITP level-2 data (which is most prominent at depth where lateral and vertical salinity changes are small), we limit our analyses from the sea surface to 400 m depth.

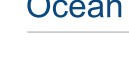 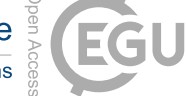

## Appendix C: Examination of the second peak in spatial autocorrelation functions

To understand the source of the second peaks found around 200 - 300 km lag in the spatial autocorrelation functions, we examine their relation to the background mean fields. The second peaks in the autocorrelation functions are always found where the corresponding $T$ and/or $S$ fields exhibit the classic circular structure associated with the anticyclonic Beaufort Gyre. Fig. C1 shows examples of the background mean fields and corresponding autocorrelation functions for various depth ranges. The upper two panels (a and c) exhibit a clear circular spatial pattern in the Canada Basin, while the lower two panels (e and g) do not. The corresponding spatial autocorrelation functions show clear second peaks around 240 km lag corresponding to the presence of the circular pattern (panels b and d), while they show no such peak where the circular pattern is not present (panels f and h).

The coincidence between the second peak and the circular structure of the Beaufort Gyre indicates that the peak captures a coherent variation of isothermal (isohaline) depth. We employ level depth surfaces for the present analysis; bowl-shaped isosurfaces of $T$ and $S$ in the Canada Basin exhibit a circular structure on level surfaces. Due to this structure, the same isothermal (isohaline) surface appears on a level surface as it encircles the center of the Beaufort gyre (Fig. C1a, c). The second peak captures a relatively high autocorrelation between the measurements, both of which belong to nearly the same isothermal (isohaline) surface but separated by a certain distance in accordance with the circular pattern. A consideration of mechanisms governing the decorrelation scale further supports this interpretation. The basin-scale dynamical response of the ocean to external forcing is manifest as vertical displacements of isopycnal surfaces (with given $T$ and $S$ properties), resulting in coherent variations of these depth surfaces. For follow on studies to the present one, it is desirable to calculate autocorrelation functions and decorrelation scales in a way that takes such coherent large-scale dynamic features into account. This could be achieved by analysing anomalies of the isohaline/isothermal's depth from their mean state. In the case of the Beaufort Gyre we expect the autocorrelation functions for the variation of the isohaline/isothermal depth to have larger spatial scales than those for $T$ and $S$ estimated on level surfaces. As an approximate measure of the decorrelation scales for isohaline/isothermal depth anomalies, we fit a Gaussian-function using the value at the zero lag correlation and the second peak obtained from the level surface analysis (Fig. C2), resulting in roughly 200 - 400 km. The largest scales we find in the 200 - 350 m depth range for the isothermal depths and in the 150 - 400 m depth range for the isohaline depths, correspond to the depths of strong vertical gradients of $T$ and $S$. For a sound analysis, a variation of iso-surface should be quantified by a variation of isosurface depth. In such an analysis, for example, salinity is no longer a variable to be examined, but depth of constant salinity surface, i.e., $Z(x, y, t)|_{S=\text{constant}}$, is the variable to be examined.



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





**Figure captions**

Figure 1: Spatial distribution of archived temperature and salinity data for the 1980 - 2015 period; a blue dot is shown if there is at least one measurement in the 0 - 20 m depth range.

Figure 2: (a) division of vertical levels for the statistical screening and decorrelation scale examination. The archived *TS* data
are classified into 50 levels according to their measurement depth. Data from an identical CTD profile are averaged over each depth range and regarded as one measurement. (b) Area mask for the area-based statistical screening and the decorrelation scale examination.

Figure 3: Temperature (left) and salinity (right) distributions versus depth (m) in the Amerasian Basin (see Figure 2b for area
definition) after a combined statistical screening. The blue dots denote data distribution after the screening, while the light-blue (green) dots denote data removed by grid-based (area-based) statistical screening. The red line denotes the mean (μ) at each depth level after the screening, and the solid and dotted black lines indicate μ±5σ and μ±10σ, respectively. The mean and standard deviations are calculated by the data from the entire Amerasian Basin.

Figure 4: Function *f* (normalized root mean square difference) of temperature (left column) and salinity (right column) in 40 - 60 m (the first row) and in 200 - 225 m (the second row) depth ranges as a function of spatial (km) and temporal (days) separations of measurement pairs. The color scale is common to the panels.

Figure 5: 0 - 90 days temporal average of the function *f* (normalized RMSD) of (a) temperature and (b) salinity as a function
of spatial separation. The thin-dashed lines denote functional form of RMSD in different depth levels, while the thick-solid black line in each panel denotes average of 0 - 400 m depth range. The thick-solid blue line is the fitted Gaussian function, $f(x) = a * exp[-(x/b)^2]$, the fitting parameters of which are shown in each panel.

Figure 6: Vertical profile of spatial scale of variation (e-folding scale of the normalized RMSD function, *f*) derived from the
fitted Gaussian function for each depth level (see also Fig. 5). The scale in each depth range is calculated from data from all seasons.

Figure 7: Background mean field of (a) temperature and (b) salinity (40 - 60 m depth range) in summer (July - September) obtained by Gaussian filtering with the e-folding scales shown in Fig. 6. A vertical filter (average of 3 adjacent layers) is
applied to the e-folding scales before the application in order to obtain a smooth transition of the filtering scale in the vertical direction. The background field is calculated only at the locations where data exist in the Amerasian Basin (bottom depth > 1000 m).





Figure 8: A summary of linear temporal trend in the Amerasian Basin: the spatial pattern of (a) temperature and (b) salinity
trend in 60 - 80 m depth range, and the time series of averaged (c) temperature and (d) salinity over the grid cells where
       trend is detected in the Amerasian Basin. The trend is calculated in each 111 km × 111 km grid cell for the period
       covered by data, and the Mann-Kendall rank statistic (*Kendal*, 1938) is applied to test the significance. In panels (a) and
       (b), only the grid cells, the trend of which are statistically significant (significance level 5%). are shown in color. Time
       series of averaged temperature/salinity over the corresponding area are shown in (c) and (d) by the thick-solid lines.
Panels (c) and (d) also exhibit averages over the grid cells, where positive (negative) trend is detected, for 350-375m
       depth by thick-solid lines, the spatial pattern of which are not shown. The dashed lines in (c) (d) depict the range of 1
       standard deviation.

Figure 9: (a) Number of data pairs used to calculate autocorrelation in each bin (log-scale) and 2-dimensional autocorrelation
function for (b) temperature and (c) salinity in 40 - 60m depth range. The color bar for (b) is common to (c). The white
       area in panels (b) and (c) indicates negative autocorrelations.

Figure 10: Spatial autocorrelation function of temperature (left column) and salinity (right column). The upper panels show
       the temporal averages of the 2-dimensional autocorrelation functions (the average of 0 - 30 days-lag  for 0 - 140 m depth
range, 0 - 60 days-lag for 140 - 400 m depth range) in various depth levels. The lower panels are Gaussian functions, the
       intercepts and e-folding scales of which are calculated from the function fitting in 0 - 150 km spatial-lag range.

Figure 11: Temporal autocorrelation function of temperature (left column) and salinity (right column). The upper panels
       show the spatial averages of the 2-dimensional autocorrelation functions (the spatial average of 0 - 20 km lag) in various
depth levels. The lower panels are Gaussian functions, the intercepts and e-folding scales of which are calculated from
       the function fitting in 0 - 200 days temporal-lag range. A 90 days temporal filter is applied to the autocorrelation
       functions in (a) and (b) to eliminate noise.

Figure 12: Vertical profiles of zero-lag autocorrelation (left column) and e-folding scale (right column) of the fitting spatial
(the first row) and temporal (the second row) autocorrelation function. A 3-layer vertical filter is applied to eliminate
       noise.

Figure 13: Vertical profile of the background mean variance, $var_{bg}$, for temperature (left) and salinity (right).

Figure B1: Vertical profiles of standard deviation of (a) temperature and (b) salinity in the Amerasian Basin. The black, blue
       and red lines indicate the standard deviation calculated from all data, ITP level-2 data, and data except ITP level-2 data,




respectively. The standard deviation at each location is calculated by the deviation from the background mean field, and then an averaged standard deviation in the entire basin is calculated.

Figure C1: Left column: examples of the background mean fields with a circular structure associated with the Beaufort Gyre (upper 2 panels) and without the circular structure (lower 2 panels). Right column: spatial autocorrelation functions corresponding to their right panels. The first, third and fourth row show temperature, while the second row shows salinity.

Figure C2: Vertical profile of the spatial decorrelation scales estimated from the second peak of the spatial autocorrelation function (see Fig. 10a, b). The scale is obtained from a Gaussian function fitting with 2 points: zero-lag autocorrelation value from Fig. 12a and the second peak. The second peak is defined by the highest autocorrelation value, the spatial lag of which is larger than 150 km. A 3-layer vertical filter is applied to eliminate noise.










Table 1. List of observational data

| Element of data compilation (alphabetical order) | Data source (URL or contact address) |
| --- | --- |
| ARAON 2011-2013 | http://eng.kopri.re.kr/home_e/ |
| ARGO 2006 − 2008 POPS | http://www.coriolis.eu.org/Data-Services-Products/View-Download |
| ARK 1993 − 2012 RV Polarstern listed in Rabe et al. (2013) | http://www.pangaea.de/ |
| ASCOS 2008 RV Oden | http://www.ascos.se |
| Beaufort Gyre Project 2003 − 2014 various ships | http://www.whoi.edu/beaufortgyre/ |
| Beringia III 2005 RV Oden | http://bolin.su.se/data/Beringia2005-Stats-Oden |
| CCGS LSSL 1997 - 2010 | Koji Shimada (koji@kaiyodai.ac.jp) |
| CCGS SWL 1999-2001, 2003, 2005-2007 | Koji Shimada (koji@kaiyodai.ac.jp) |
| CHINARE 1999 − 2010 RV Xuelong | http://www.nsfcodc.cn/polar/ |
| CLIVAR and Carbon Hydrographic data Office (CBL02, Oden91, PK-ARK-XII, SBI03) | http://cchdo.ucsd.edu/arctic |
| DAMOCLES 2006 − 2008 POPS | http://www.ipev.fr/damocles/ |
| Greenland Sea Project (1987-1993) | http://ocean.ices.dk/Project/GSP/ |
| ICES datasets (CTD and bottle data, 1980-2015) | http://ices.dk/marine-data/dataset-collections/Pages/default.aspx |
| ITP level-3 data (1-19, 21-23, 25-28, 30, 32, 33, 35, 36, 41, 42) | http://www.whoi.edu/itp |
| ITP level-2 data (24, 29, 31, 34, 37-40, 43-94)  update Dec. 03, 2015 | http://www.whoi.edu/itp |
| JAMSTEC 1999 − 2010, 2012 RV Mirai | http://www.godac.jamstec.go.jp/darwin/datatree/e |
| JAMSTEC Compact Arctic Drifter 5, 6 | http://psc.apl.washington.edu/northpole/Data.html |
| Larsen 93 cruise | Koji Shimada (koji@kaiyodai.ac.jp) |
| LOMROG 2007 2007 RV Oden NODC OAS accession 0093533 | http://www.nodc.noaa.gov/cgi-bin/OAS/prd/accession/0093533 |
| N/A 2001 RV Oden NODC OAS accession 0002194 | http://www.nodc.noaa.gov/cgi-bin/OAS/prd/accession/0002194 |
| NABOS/CABOS data (2002 − 2009, 2013 and 2015) | http://nabos.iarc.uaf.edu/ |
| NPEO 2000 − 2014 airborne and ice-based | ftp://psc.apl.washington.edu/NPEO Data Archive/NPEO Aerial CTDs/ |
| PAICEX 2007 − 2009ice-based | Sergey Pisarev (pisarev@ocean.ru) |
| PANGAEA (POMAR, Yakov Simmitsky, LANCE cruises) | http://www.pangaea.de/ |
| PS86 & PS87 XCTD | http://www.pangaea.de/ |
| SCICEX 1993 US submarines and ice-based | http://data.eol.ucar.edu/codiac/dss/id=106.arcss072/ |
| SCICEX 1996-1999, US submarines and ice-based SAIC project | Sergey Pisarev (pisarev@ocean.ru) |
| SCICEX 1997 and 1998, US submarines and ice-based | http://data.eol.ucar.edu/codiac/dss/id=106.arcss064/ |
| SCICEX 2000 US submarine | ftp://sidads.colorado.edu/pub/DATASETS/NOAA/G02187/XCTD/2000/edffiles/ |
| SCICEX 2001 US submarine | http://www.nodc.noaa.gov/archive/arc0021/0000568/1.1/data/0-data/Scranton-01/Probe Data/English EDFs/ |
| SCICEX 2003 US submarine | ftp://sidads.colorado.edu/pub/DATASETS/NOAA/G02187/XCTD/2003/edffiles/ |
| SCICEX 2014 US submarine | ftp://sidads.colorado.edu/pub/DATASETS/NOAA/G02187/XCTD/2014/uss-new-mexico/ |
| Switchyard 2003 − 2012 ice-based | http://data.eol.ucar.edu/codiac/dss/id=106.ARCSS129 |
| System Laptev Sea Project, 2007 - 2011 | Markus Janout (Markus.Janout@awi.de),  Jens Hölemann (Jens.Hoelemann@awi.de) |
| UNCLOS 2011 CCGS LSSL | Takashi Kikuchi (takashik@jamstec.go.jp) |
| WOD13 (APB, CTD, DRB, GLD, MRB, OSD, PFL, SUR, UOR; 1980-2013) | https://www.nodc.noaa.gov/OC5/WOD13/ |



Figure 1: Spatial distribution of archived temperature and salinity data for the 1980 - 2015 period; a blue dot is shown if there is at least one measurement in the 0 - 20 m depth range.





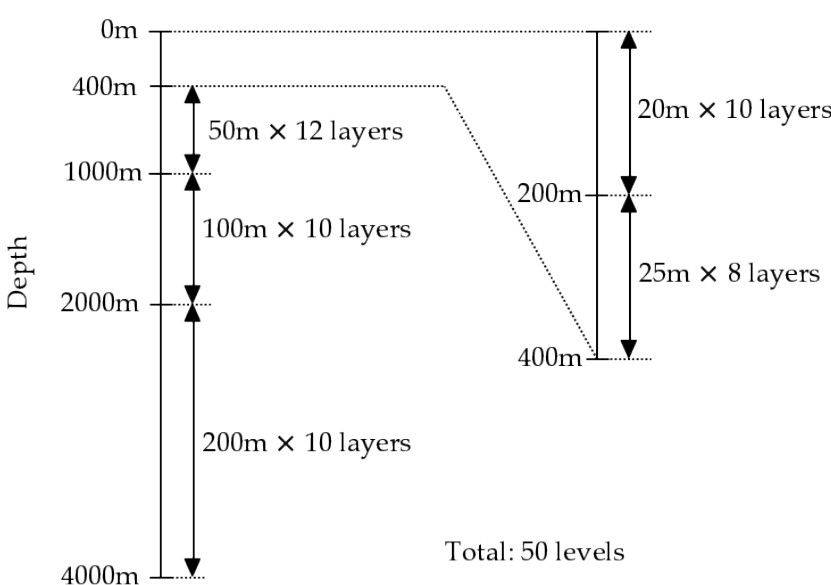

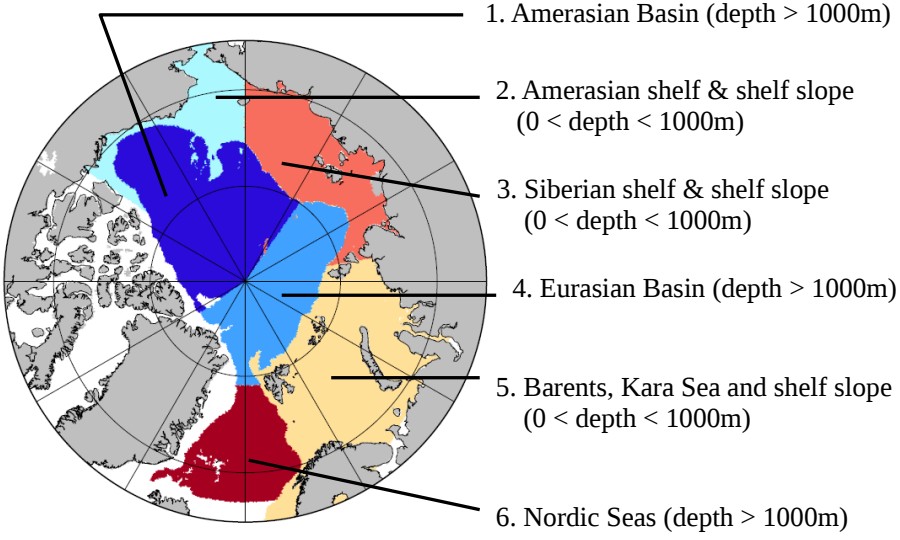

Figure 2: (a) division of vertical levels for the statistical screening and decorrelation scale examination. The archived *TS* data are classified into 50 levels according to their measurement depth. Data from an identical CTD profile are averaged over each depth range and regarded as one measurement. (b) Area mask for the area-based statistical screening and the decorrelation scale examination.





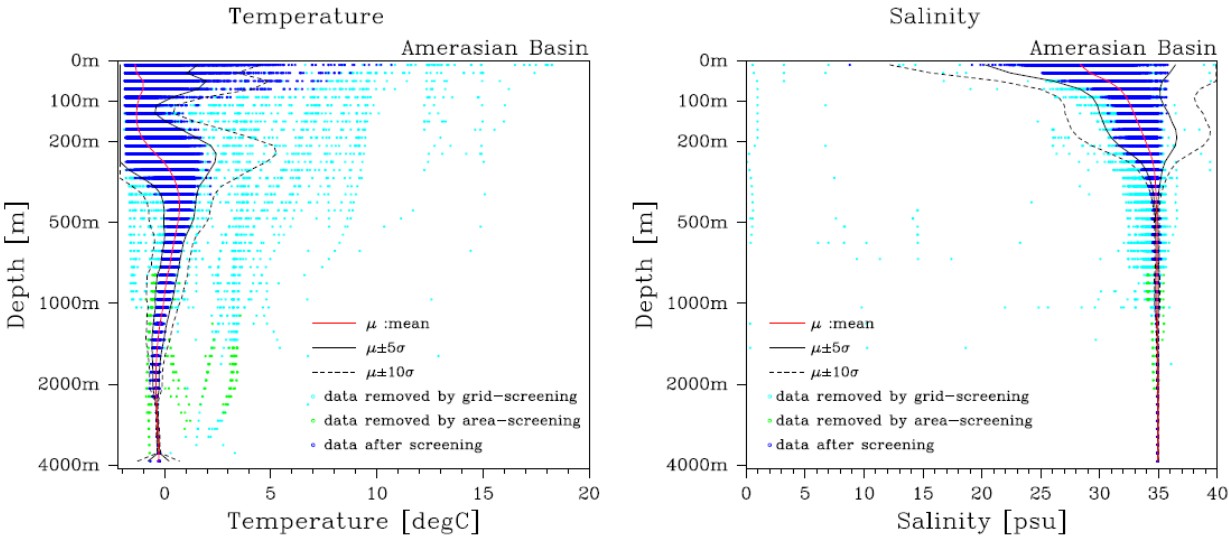

Figure 3: Temperature (left) and salinity (right) distributions versus depth (m) in the Amerasian Basin (see Figure 2b for area definition) after a combined statistical screening. The blue dots denote data distribution after the screening, while the light-blue (green) dots denote data removed by grid-based (area-based) statistical screening. The red line denotes the mean (μ) at each depth level after the screening, and the solid and dotted black lines indicate μ±5σ and μ±10σ, respectively. The mean and standard deviations are calculated by the data from the entire Amerasian Basin.



Figure 4: Function *f* (normalized root mean square difference) of temperature (left column) and salinity (right column) in 40 - 60 m (the first row) and in 200 - 225 m (the second row) depth ranges as a function of spatial (km) and temporal (days) separations of measurement pairs. The color scale is common to the panels.





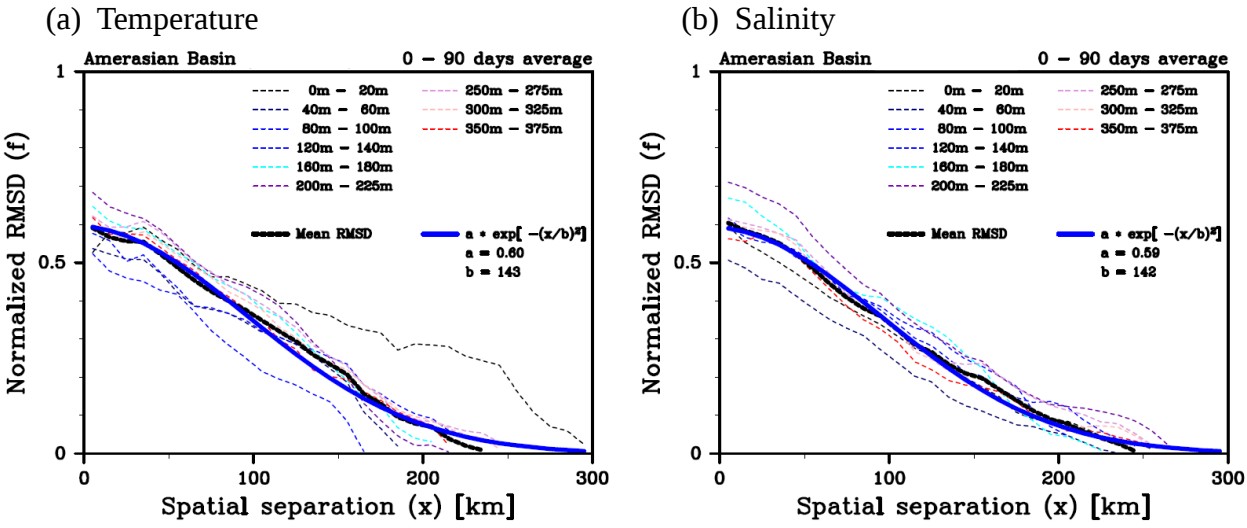

Figure 5: 0 - 90 days temporal average of the function *f* (normalized RMSD) of (a) temperature and (b) salinity as a function of spatial separation. The thin-dashed lines denote functional form of RMSD in different depth levels, while the thick-solid black line in each panel denotes average of 0 - 400 m depth range. The thick-solid blue line is the fitted Gaussian function, $f(x) = a * \exp[-(x/b)^2]$, the fitting parameters of which are shown in each panel.





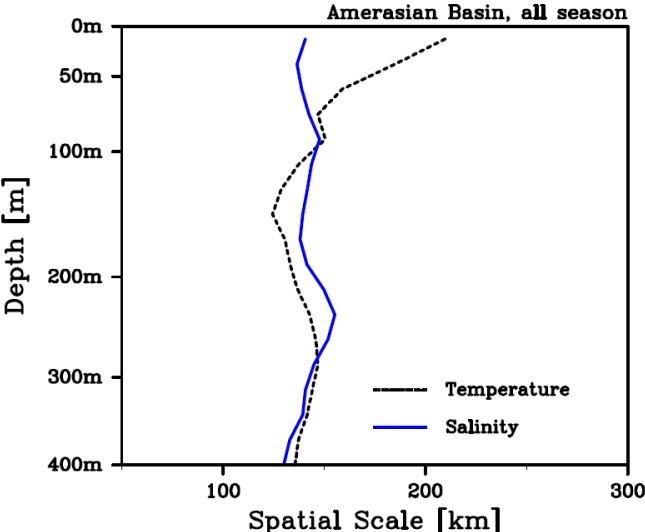

Figure 6: Vertical profile of spatial scale of variation (e-folding scale of the normalized RMSD function, *f*) derived from the fitted Gaussian function for each depth level (see also Fig. 5). The scale in each depth range is calculated from data from all seasons.




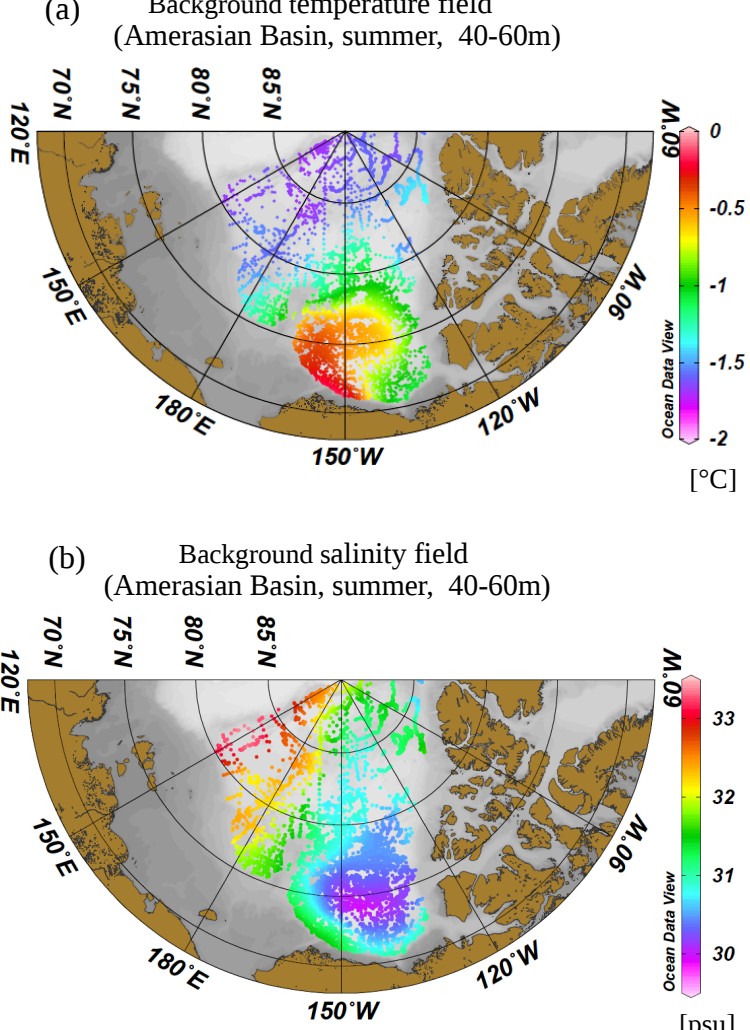

Figure 7: Background mean field of (a) temperature and (b) salinity (40 - 60 m depth range) in summer (July - September) obtained by Gaussian filtering with the e-folding scales shown in Fig. 6. A vertical filter (average of 3 adjacent layers) is applied to the e-folding scales before the application in order to obtain a smooth transition of the filtering scale in the vertical direction. The background field is calculated only at the locations where data exist in the Amerasian Basin (bottom depth > 1000 m).


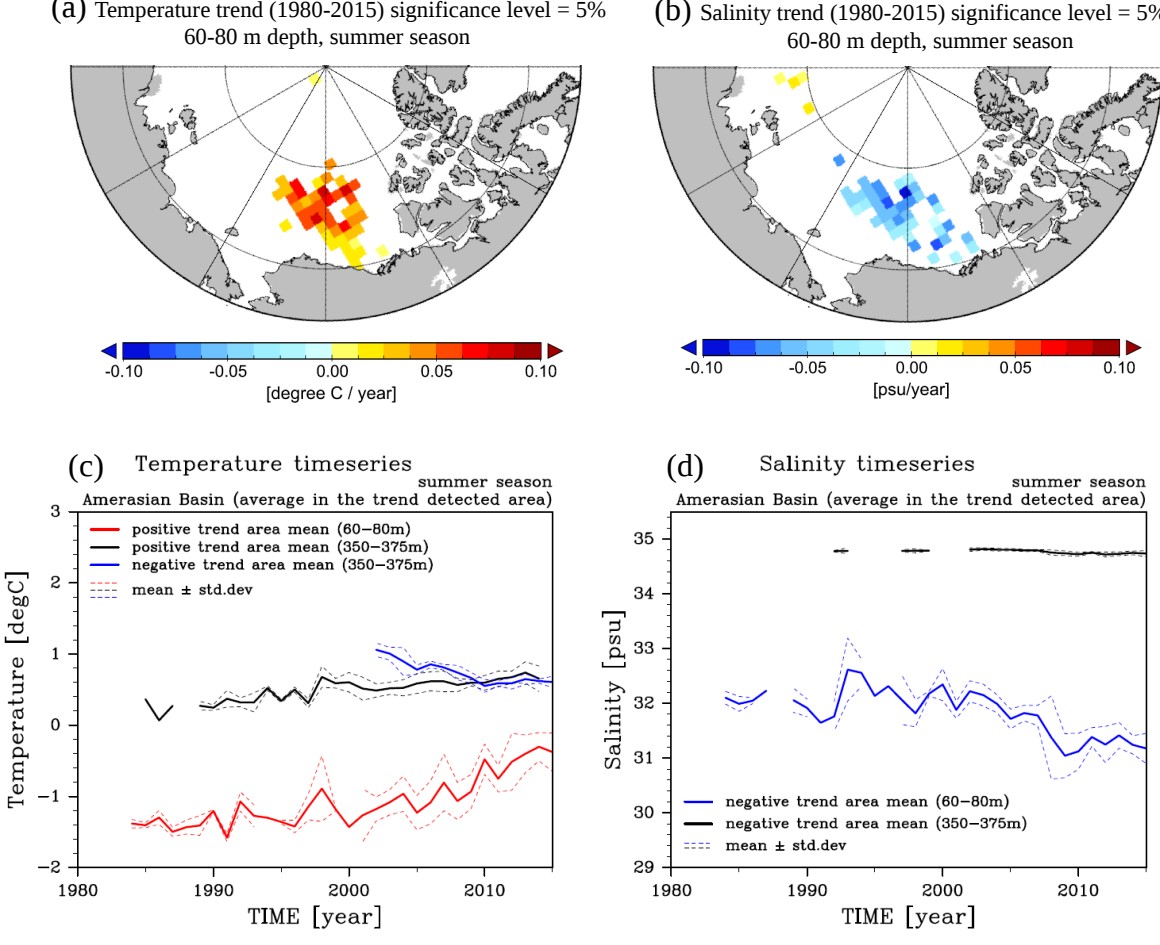

Figure 8: A summary of linear temporal trend in the Amerasian Basin: the spatial pattern of (a) temperature and (b) salinity trend in 60 - 80 m depth range, and the time series of averaged (c) temperature and (d) salinity over the grid cells where trend is detected in the Amerasian Basin. The trend is calculated in each 111 km × 111 km grid cell for the period covered by data, and the Mann-Kendall rank statistic (*Kendal*, 1938) is applied to test the significance. In panels (a) and (b), only the grid cells, the trend of which are statistically significant (significance level 5%). are shown in color. Time series of averaged temperature/salinity over the corresponding area are shown in (c) and (d) by the thick-solid lines. Panels (c) and (d) also exhibit averages over the grid cells, where positive (negative) trend is detected, for 350-375m depth by thick-solid lines, the spatial pattern of which are not shown. The dashed lines in (c) (d) depict the range of 1 standard deviation.



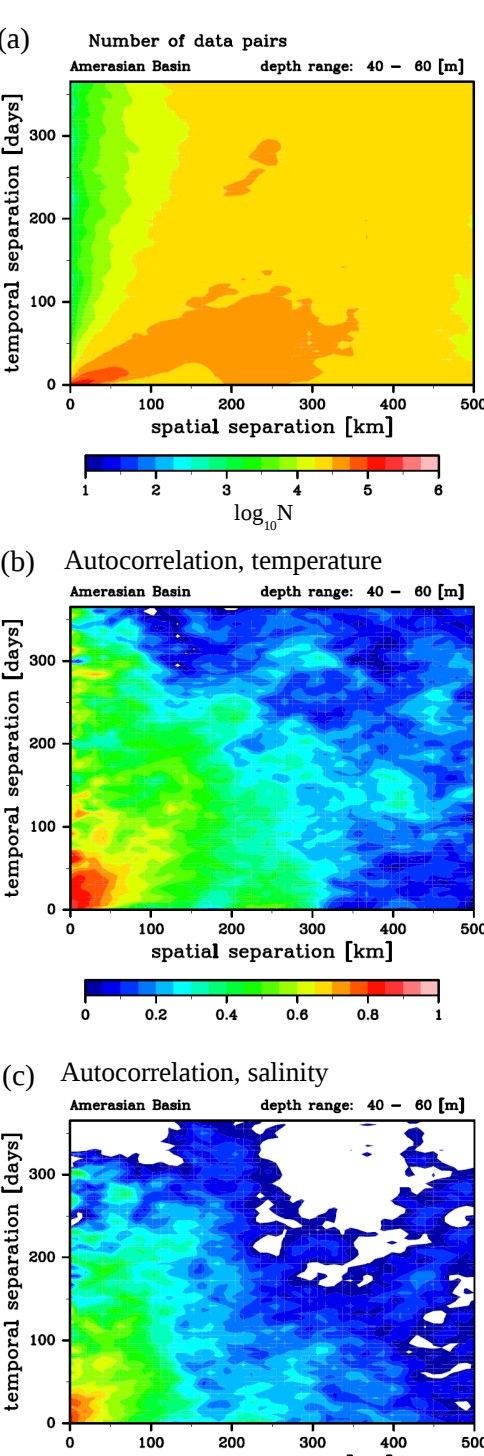

Figure 9: (a) Number of data pairs used to calculate autocorrelation in each bin (log-scale) and 2-dimensional autocorrelation function for (b) temperature and (c) salinity in 40 - 60m depth range. The color bar for (b) is common to (c). The white area in panels (b) and (c) indicates negative autocorrelations.





Figure 10: Spatial autocorrelation function of temperature (left column) and salinity (right column). The upper panels show the temporal averages of the 2-dimensional autocorrelation functions (the average of 0 - 30 days-lag for 0 - 140 m depth range, 0 - 60 days-lag for 140 - 400 m depth range) in various depth levels. The lower panels are Gaussian functions, the intercepts and e-folding scales of which are calculated from the function fitting in 0 - 150 km spatial-lag range.

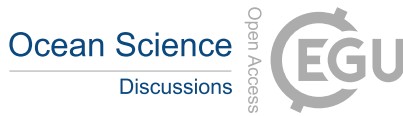

Figure 11: Temporal autocorrelation function of temperature (left column) and salinity (right column). The upper panels show the spatial averages of the 2-dimensional autocorrelation functions (the spatial average of 0 - 20 km lag) in various depth levels. The lower panels are Gaussian functions, the intercepts and e-folding scales of which are calculated from the function fitting in 0 - 200 days temporal-lag range. A 90 days temporal filter is applied to the autocorrelation functions in (a) and (b) to eliminate noise.




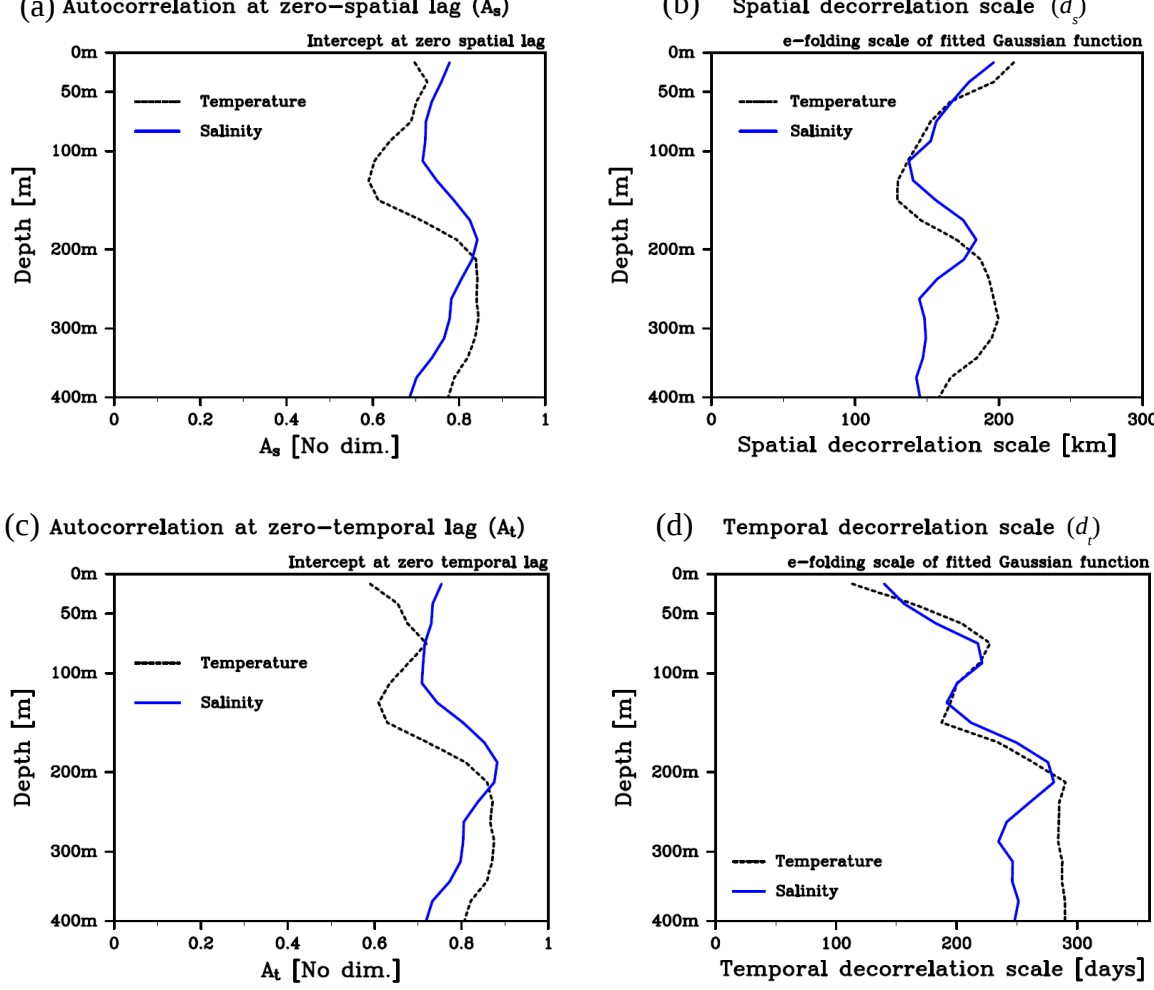

Figure 12: Vertical profiles of zero-lag autocorrelation (left column) and e-folding scale (right column) of the fitting spatial (the first row) and temporal (the second row) autocorrelation function. A 3-layer vertical filter is applied to eliminate noise.





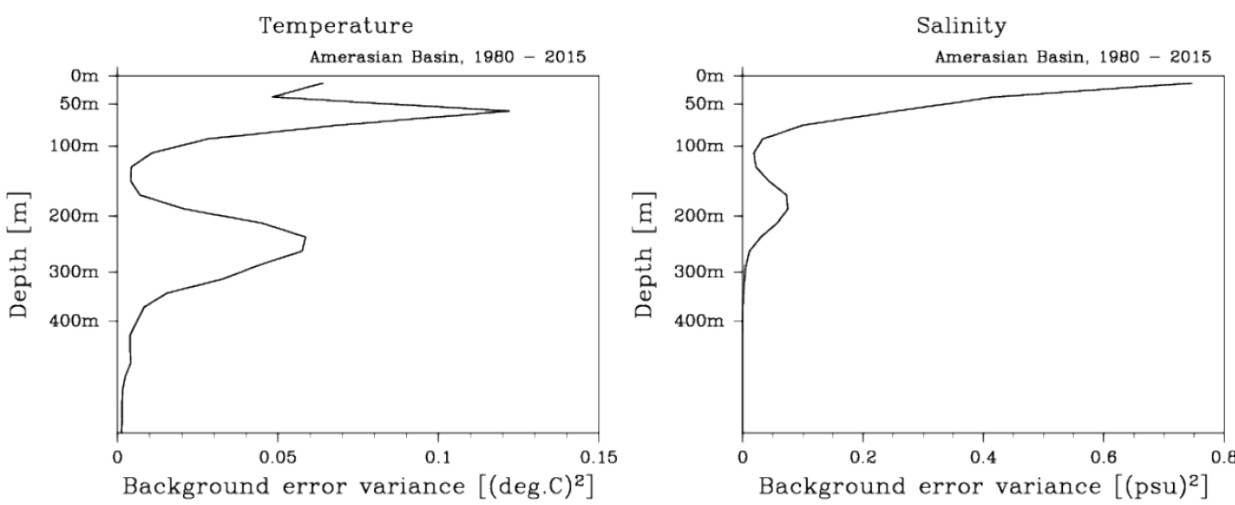

Figure 13: Vertical profiles of the background mean variance, $var_{bg}$, for temperature (left) and salinity (right).





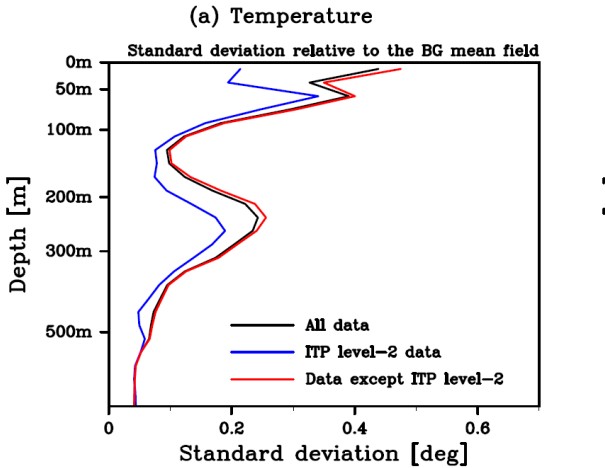
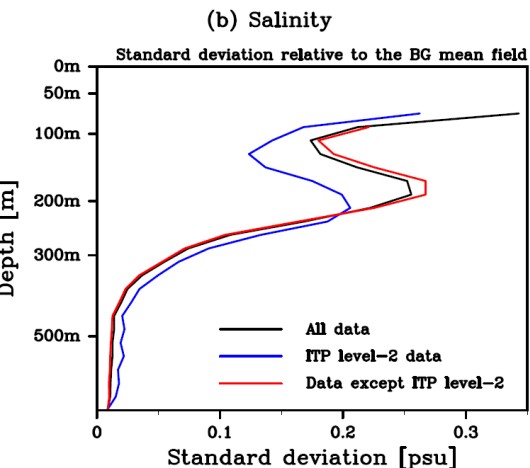

Figure B1: Vertical profiles of standard deviation of (a) temperature and (b) salinity in the Amerasian Basin. The black, blue and red lines indicate the standard deviation calculated from all data, ITP level-2 data, and data except ITP level-2 data, respectively. The standard deviation at each location is calculated by the deviation from the background mean field, and then an averaged standard deviation in the entire basin is calculated.





Figure C1: Left column: examples of the background mean fields with a circular structure associated with the Beaufort Gyre (upper 2 panels) and without the circular structure (lower 2 panels). Right column: spatial autocorrelation functions corresponding to their right panels. The first, third and fourth row show temperature, while the second row shows salinity.





Figure C2: Vertical profile of the spatial decorrelation scales estimated from the second peak of the spatial autocorrelation function (see Fig. 10a, b). The scale is obtained from a Gaussian function fitting with 2 points: zero-lag autocorrelation value from Fig. 12a and the second peak. The second peak is defined by the highest autocorrelation value, the spatial lag of which is larger than 150 km. A 3-layer vertical filter is applied to eliminate noise.