# Peer review of "Decorrelation scales for Arctic Ocean Hydrography. Part I: Amerasian Basin."

_Ocean Science, 2017_

## Referee Comment (RC1) · Anonymous Referee #1 · 4 Oct 2017

Review of Sumata et al., "Decorrelation scales for Arctic Ocean hydrography. Part 1: Amerasian Basin," submitted to Ocean Science Discussions.

This is a very nice paper, and very well written. I have some minor comments that should be addressed before publication.

Line 66: You should define "representation error."

Line 120: Here you state that the vertical depth range of the analysis is 0-400 m. Then in Section 2.3, Figure 2, and beyond, you discuss depths deeper than 400 m. This is confusing.

Line 151: Please explain the "vertical stability test" in more detail.

[Figure]

Equation 1: It is a very poor choice of variable names to use T and S for "observed property" and "structure function" respectively, considering the key roles here of T and S for temperature and salinity. I strongly suggest a change in variable names.

Line 269: 10 km interval, 10 day interval. There are many instances (this is just one) where you have chosen a length or time scale for analysis, for binning, for smoothing, etc. A few times you justify your choice, but most of the time you do not, as in this instance. I suggest that each time you introduce a space or time scale, you provide some kind of justification for these choices, or an analysis that shows that the results are insensitive to the choice. (Another example: 10 km interval, 5-day interval, Line 379.)

Figure 8 and first paragraph of Section 3.3: Your text mentions 0-160 m depths, yet the figure shows 60-80 m; also text 200-400 m, but figure 350-375 m. This is confusing. Further, the black line evidently corresponds to the "southern perimeter of the Canada Basin" but this is not obvious in the figure nor is it explained as such in the figure caption.

Lines 337-339: Are you simply stating here the obvious point that real long-term trends might not be accurately represented by your sparse data set? This hardly needs mentioning. Or are you trying to make a different point? Please clarify.

Lines 341-345: It might be very interesting to see a map of the representative time.

Figure 9 and associated text: I think you should provide some discussion comparing Figure 9 and Figure 4: Why are they so similar, ie why is the scale of the mean field similar to the scale of the variance?

Lines 426-436: This is very interesting but it is speculative and should be written as such. IE I suggest changing text such as "due to . . ." to "possibly due to . . ."

Figure 13: Your text provides a brief explanation of the figure, but then provides no analysis. This suggests to me that it is not necessary. If you disagree, then I suggest

that you provide some analysis.

Appendix B: Is there an error in the Figure B1? It shows only a tiny bit more variance in ITP level-2 salinity at depth, and no more in temperature, in contrast to your claim in the text.

Figure 1: 1) Use a different color for the bathymetry vs. the data dots. 2) Please provide a bathymetric color scale. 3) Please provide geographic place names used in the paper on this figure.

Figure 3 and others: It is very difficult or impossible to see the difference in color between tiny colored dots or lines.

---

## Referee Comment (RC2) · Anonymous Referee #2 · 11 Dec 2017

This is a nice manuscript, clearly presented and largely well-written. It is a technical piece on decorrelation scales in the Amerasian Basin of the Arctic Ocean, and as such, it's maybe not fascinating, but it will be useful, and I'm happy to recommend it for publication.

I cannot help but notice that the other review has trapped the queries that I had (rather more, in fact), so I will not repeat them here.
* * *

---

## Author Comment (AC1) · 21 Dec 2017

Reply to the review comments by anonymous reviewer #1.

We appreciate comments and suggestions from the anonymous reviewer.
In the following text, the comments from the reviewer are shown in black while our reply is written in blue . The line numbers shown in our reply refer those in the revised "track changes" manuscript.

Review of Sumata et al., "Decorrelation scales for Arctic Ocean hydrography. Part 1:
Amerasian Basin," submitted to Ocean Science Discussions.
This is a very nice paper, and very well written. I have some minor comments that should be addressed before publication.

Line 66: You should define "representation error."

Thank you for the comment.
We revised the manuscript to clarify the definition of "representation error" as follows;
"For a model - data misfit calculation, the difference of the spatial (and temporal) scales represented by a model and by the observations should be taken into account. Physical properties simulated in General Circulation Models (GCMs) represent mean values over each grid cell for a certain temporal period, whereas those from in-situ measurements represent values at a localized point in space and in time. The error resulting from the difference of the scales represented by these two approaches is referred to as representation error (see van Leeuwen [2015] for a summary). The autocorrelation function and the decorrelation scales provide a direct measure of the representation error." (line 64-71 in the revised manuscript).

Line 120: Here you state that the vertical depth range of the analysis is 0-400 m. Then in Section 2.3, Figure 2, and beyond, you discuss depths deeper than 400 m. This is confusing.

Although we presented an analysis for the depth range 0 - 400 m for the Amerasian Basin in this manuscript, the quality control procedure described in sec. 2 will be also applied to other parts of the Arctic Ocean in a forthcoming paper. We would like to keep the description as it is, but we revised the manuscript to avoid confusion. In the revised manuscript we added;
"Note that although we describe the QC procedure as it is applied to the entire raw data set in this section, we will use only data from 0 - 400 m depth (after the QC) in the present scale analysis as mentioned in the introduction." (line 182-184)
We also provided a notification in the caption of Fig. 3 as follows;
"Note that analyses in sec. 3 and 4 use data from 0 – 400 m depth range." (line 932-933)

Line 151: Please explain the "vertical stability test" in more detail.

The vertical stability test is applied to remove spike data points found in CTD and XCTD profiles. The quality control algorithm compares density between vertically neighbouring points in each profile. If the program finds a vertical density inversion, both of the data points are removed from the profile data. The rest of the data points in the profile are still used in the present analyses (if they pass other QC criteria). We provided an additional sentence to mention this procedure (line 160-162).

Equation 1: It is a very poor choice of variable names to use T and S for "observed property" and "structure function" respectively, considering the key roles here of T and S for temperature and salinity. I strongly suggest a change in variable names.

Thank you for the suggestion. We agree the point and revised the manuscript.
In the revised manuscript, capital omega ($\Omega$) for observed property and phi ($\varphi$) for the structure function are used.

Line 269: 10 km interval, 10 day interval. There are many instances (this is just one) where you have chosen a length or time scale for analysis, for binning, for smoothing, etc. A few times you justify your choice, but most of the time you do not, as in this instance. I suggest that each time you introduce a space or time scale, you provide some kind of justification for these choices, or an analysis that shows that the results are insensitive to the choice. (Another example: 10 km interval, 5-day interval, Line 379.)

We provided explanations and/or reasoning when we introduce binning scales as follows;
- Binning separation used for the analysis of spatial scale of variation: "For the binning we suppose that the spatial and temporal scales of variation are much larger than the scale used for the binning, the validity of which is recursively confirmed by the scales estimated." (line 281-283)
- Size of the grid cells used for the trend analysis: "The size of the grid cells is chosen to be consistent with the spatial scale of variation (sec. 3.2)" (line 337-338)

- Binning separation for autocorrelation function analysis: "The spatial and temporal sizes of the bin are designed to capture the functional form of the autocorrelation relevant for basin-scale data assimilation (i.e., the functional form of the autocorrelation describing mesoscale fluctuations are not examined in this analysis)." (line 399-401)

Figure 8 and first paragraph of Section 3.3: Your text mentions 0-160 m depths, yet the figure shows 60-80 m; also text 200-400 m, but figure 350-375 m. This is confusing. Further, the black line evidently corresponds to the "southern perimeter of the Canada Basin" but this is not obvious in the figure nor is it explained as such in the figure caption.

Since we analyze the data in vertically discretized bins, the time series in Figure 8 (60-80 m bin and 350-375 m bin) are shown as examples representing common features observed in the 0-160 m and 200-400 m depth ranges. To clarify this point, we revised the manuscript as follows;
"A positive trend in $T$ is observed in the depth range from 0 m to 160 m over the whole analysed time period (i.e., through the Pacific-water/upper halocline layers, represented by red line in Fig. 8c), while after the year 2002 a decreasing trend in $T$ is observed in the central Canada Basin in the 200 - 400 m depth range (lower halocline/Atlantic-water layer, represented by blue line in Fig. 8c). A positive trend is observed along the southern perimeter of the Canada Basin in 250 - 400 m depth range (Atlantic-water layer, represented by black line in Fig. 8c and d)." (line 342-350)
For the black line shown in Figure 8 (c) and (d), we provided further explanations in the figure caption;
"Black thick solid lines in panels (c) and (d) exhibit averages over the grid cells, where positive (negative) trends of $T$ ($S$) are detected along the southern perimeter of the Canada Basin in the 350-375m depth range (spatial pattern is not shown)." (line 960-963)

Lines 337-339: Are you simply stating here the obvious point that real long-term trends might not be accurately represented by your sparse data set? This hardly needs mentioning. Or are you trying to make a different point? Please clarify.

We removed the sentence.

Lines 341-345: It might be very interesting to see a map of the representative time.

We added a map of the representative time for each season in the supplementary material.

Figure 9 and associated text: I think you should provide some discussion comparing Figure 9 and Figure 4: Why are they so similar, ie why is the scale of the mean field similar to the scale of the variance?

Although we agree that this is an interesting question which should be addressed, we cannot provide a good explanation or interpretation. We expect that the the scales of the mean field and the anomalies are different for other basins (e.g. interior basin, over the continental shelf and over the shelf slope). Since we plan to analyze decorrelation scales in the Eurasian basin and over the Arctic shelf slope in a forthcoming paper, we will revisit the question in future.
We added the following sentence to address the reviewer's concern (Fig. 6 and Fig. 12b are compared since they summarize the scales);
"It is interesting to note that the scales of the mean field and of the variance are very similar (e.g., cmp. Fig. 6 and Fig. 12b). We currently have no explanation for this feature but assume that it is a peculiarity based on the dynamics of the analyzed basin. In forthcoming papers we plan to analyze the scales in the Eurasian basin and over the Arctic shelf slope and will revisit this question." (line 457-460)

Lines 426-436: This is very interesting but it is speculative and should be written as such. IE I suggest changing text such as "due to . . ." to "possibly due to . . ."

We agree the point and revised the manuscript as the reviewer suggested (line 455-457).

Figure 13: Your text provides a brief explanation of the figure, but then provides no analysis. This suggests to me that it is not necessary. If you disagree, then I suggest that you provide some analysis.

Thank you for the comments. We think further analysis of this figure is not necessary, while we think it is meaningful to provide interpretation of these profiles. To clarify the relation between the vertical profile of background mean variance (Figure 13) and Amerasian Basin hydrography, we added the following paragraph in section 4.3;
"The background mean variance clearly reflects the vertical stratification in the Amerasian Basin [e.g., McLaughlin et al., 2004, Shimada et al., 2005], with highest variance in the depth ranges of vertical extrema in the profile. The temperature profile exhibits two minima (in the mixed layer and around 130 m depth) and two maxima (approximately in 70 m and 250 m), Figure 3 (left). These shallow extrema are associated with the seasonally, spatially and interannually varying Near Surface Temperature Maximum (see e.g., McPhee et al., 1998), and Pacific Summer Water layers (see e.g., Timmermans et al., 2014). The deep minimum correspond to the Pacific Winter Water layer plus

variations in the deeper Atlantic Water (see e.g., Shimada et al., [2005] Fig. 2). The vertical profile of salinity variance also exhibits good correspondence with salinity stratification and its variation (Fig. 3, right), with smallest variance (approximately 120 m depth) corresponding to weakest salinity stratification and largest (around 180 m) corresponding to the stratification boundary between the upper and lower halocline. The derived covariance is also necessary to complete the model - observation misfit calculation, as summarized in the following section." (line 484-494)

Appendix B: Is there an error in the Figure B1? It shows only a tiny bit more variance in ITP level-2 salinity at depth, and no more in temperature, in contrast to your claim in the text.

There is no error in Figure B1, although our description in the text might be ambiguous and/or insufficient. As pointed out by the reviewer, the figure shows only a tiny bit more variance in ITP level-2 salinity at depth. Nevertheless, the excess of the variance is comparable to the magnitude of natural variability in this depth range. To clarify the point, we rewrote the second paragraph of Appendix B as follows;
"As a measure of the uncertainty of the uncalibrated ITP level-2 data, we calculate deviations of the ITP level-2 data from the background mean field (sec 3.2). We assume that the standard deviations of the background field derived from all data represent natural variability of $T$ and $S$ in each depth level. If the standard deviation from ITP level-2 data is larger than the natural variability, we can conclude that the ITP level-2 data has an error (bias) expressed by the excess of the standard deviation. Fig. B1 depicts vertical profiles of the standard deviations of $T$ and $S$ calculated from all data, from ITP level-2 data only, and from all data except ITP level-2 data. The $T$ profiles exhibit smaller standard deviation of ITP level-2 data than the natural variability throughout the entire water column. On the other hand, the $S$ profile shows that the standard deviation of ITP level-2 data is larger than the natural variability below 250 m depth, and it is almost double as large below 500 m depth.  Since the spatial scale estimated in sec. 3.1 and the decorrelation scale estimated in sec. 4.2 would be deteriorated by erroneous sensor drifts, we limit our analyses from the sea surface to 400 m depth." (line 632-647)

Figure 1: 1) Use a different color for the bathymetry vs. the data dots. 2) Please provide a bathymetric color scale. 3) Please provide geographic place names used in the paper on this figure.

Thank you for the suggestion. We redrew Figure 1.
In the revised figure we provide two panels; one contains bathymetric information with color bar and geographic names used in this manuscript. The other panel shows the location of observation with a different color as the background topography.

Figure 3 and others: It is very difficult or impossible to see the difference in color between tiny colored dots or lines.

We redrew Figure 1 and 3 with different color for dots to make the contrast clearer, and with increased resolution.

[revised manuscript text omitted]

**Representative year in each 111 x 111 km grid cell**

[Figure]

Winter (Jan. - Mar.)

Spring (Apr. - Jun)

Summer (Jul. - Sep.)

Autumn (Oct. - Dec.)

representative_year (lev = 01, season = 1)

representative_year (lev = 01, season = 2)

representative_year (lev = 01, season = 3)

representative_year (lev = 01, season = 4)

Representative year of the data in each grid cell used as the tie point for the time-varying background mean field.

---

## Author Comment (AC2) · 21 Dec 2017

Reply to the review comments by anonymous reviewer #2.

We appreciate the anonymous reviewer #2 for reviewing our manuscript entitled "Decorrelation scales for Arctic Ocean Hydrography. Part I: Amerasian Basin".
Since the reviewer #2 did not raise additional points but assigned to refer the other review, we suppose that our point-by-point reply to the other review and the revised manuscript could provide response to (supposed) queries by reviewer #2.

Comments from the reviewer #2:
This is a nice manuscript, clearly presented and largely well-written. It is a technical
piece on decorrelation scales in the Amerasian Basin of the Arctic Ocean, and as
such, it's maybe not fascinating, but it will be useful, and I'm happy to recommend it for
publication.
I cannot help but notice that the other review has trapped the queries that I had (rather
more, in fact), so I will not repeat them here.

Thank you again for the review on our manuscript.